# FairNet: Dynamic Fairness Correction without Performance Loss via Contrastive Conditional LoRA

Songqi Zhou          Zeyuan Liu          Benben Jiang*

**Department of Automation**
Tsinghua University
Beijing, China
zhousongqi@tsinghua.edu.cn, liuzeyuan23@mails.tsinghua.edu.cn,
bbjiang@tsinghua.edu.cn

## Abstract

Ensuring fairness in machine learning models is a critical challenge. Existing debiasing methods often compromise performance, rely on static correction strategies, and struggle with data sparsity, particularly within minority groups. Furthermore, their utilization of sensitive attributes is often suboptimal, either depending excessively on complete attribute labeling or disregarding these attributes entirely. To overcome these limitations, we propose FairNet, a novel framework for dynamic, instance-level fairness correction. FairNet integrates a bias detector with conditional low-rank adaptation (LoRA), which enables selective activation of the fairness correction mechanism exclusively for instances identified as biased, and thereby preserve performance on unbiased instances. A key contribution is a new contrastive loss function for training the LoRA module, specifically designed to minimize intra-class representation disparities across different sensitive groups and effectively address underfitting in minority groups. The FairNet framework can flexibly handle scenarios with complete, partial, or entirely absent sensitive attribute labels. Theoretical analysis confirms that, under moderate TPR/FPR for the bias detector, FairNet can enhance the performance of the worst group without diminishing overall model performance, and potentially yield slight performance improvements. Comprehensive empirical evaluations across diverse vision and language benchmarks validate the effectiveness of FairNet. Code is available at https://github.com/SongqiZhou/FairNet.

## 1 Introduction

The widespread implementation of machine learning (ML) in high-stakes domains such as finance, hiring, criminal justice, and healthcare [23, 9], while promisingly increasing efficiency, has sparked significant concerns regarding fairness. A primary issue is the tendency of ML models to inherit and potentially exacerbate latent social biases within training data [1, 2, 27], thus leading to discriminatory outcomes against protected groups (e.g., based on race or gender). Evidence of this includes automated financial systems that disproportionately deny creditworthy minority applicants [8], hiring tools that perpetuate gender stratification [4], and diagnostic systems that produce lower accuracy for underrepresented patient populations [28]. These systematic biases erode trust in AI systems and pose considerable ethical and legal challenges [6, 29]. Consequently, confronting and mitigating algorithmic bias is no longer a solely technical pursuit, but an urgent societal imperative.

---

*Corresponding author.

39th Conference on Neural Information Processing Systems (NeurIPS 2025).

However, devising effective debiasing techniques remains a challenge due to several persistent obstacles that limit their real-world efficacy. A major impediment is the well-documented 'performance-fairness trade-off' [32, 14, 30], where interventions aimed at fairness frequently diminish overall model performance (e.g. accuracy), thus limiting deployment. Additionally, current paradigms for bias mitigation, including pre-processing [37, 3], in-processing [17, 33], and post-processing [16, 35], predominantly rely on static global adjustments. Such 'one-size-fits-all' interventions disregard the instance-level nuances of bias, risking suboptimal outcomes like over-correction for some samples and under-correction for others. Compounding this, models trained via standard Empirical Risk Minimization (ERM) often struggle with data sparsity in minority subgroups, leading to underfitting and disproportionately high error rates for these populations [11, 21, 38]. Finally, existing methods lack flexibility in leveraging sensitive attributes: many demand complete annotations, which are often impractical or costly to obtain, while others entirely discard potentially valuable partial label information, thus limiting the potential for more targeted and effective bias mitigation [12, 22].

To address these limitations, we propose FairNet, a novel framework for dynamic, instance-level fairness correction operating internally within the model. Unlike conventional static approaches, FairNet employs an internal, lightweight bias detector that analyzes instance representations during inference. Based on this detection, parameter-efficient Low-Rank Adaptation (LoRA) modules [13] are conditionally activated to apply targeted, corrective adjustments directly at the representation level, enabling nuanced bias mitigation. These adjustments are applied selectively to instance representations identified as potentially biased, preserving performance on others and thus addressing the performance-fairness trade-off. We train the corrective LoRA module using a novel contrastive loss specifically designed to minimize intra-class representation gaps between sensitive attribute groups, thereby tackling minority subgroup underfitting. Furthermore, FairNet is architected for flexibility, effectively leveraging full, partial, or even absent sensitive attribute labels.

Complementing the framework's design, our theoretical analysis provides formal performance guarantees. Specifically, we establish that under moderate and practically achievable conditions on the bias detector's true positive rate (TPR) and false positive rate (FPR), FairNet can provably enhance performance on the worst-performing subgroup without sacrificing overall performance , and may even yield marginal gains. Our main contributions are:

1. **A novel framework, FairNet,** enabling dynamic, instance-level fairness correction via conditional LoRA, offering a new paradigm for selective bias mitigation operating directly within model representations.

2. **A targeted contrastive loss function** specifically designed to minimize intra-class representation discrepancies across sensitive groups, effectively mitigating the common issue of minority subgroup underfitting at the representation level.

3. **An adaptable framework design** demonstrating effectiveness and flexibility across varying levels of sensitive attribute label availability, including scenarios with full, partial, or entirely absent labels, enhancing practical applicability.

4. **Theoretical guarantees coupled with comprehensive empirical validation** across diverse vision and language benchmarks, demonstrating FairNet's ability to enhance worst-group accuracy without sacrificing overall performance.

## 2 Related Work

### 2.1 Diversity and Challenges of Fairness Definitions and Metrics

Fairness in machine learning is predominantly investigated within two paradigms: *group fairness* and *individual fairness*. Group fairness aims to achieve statistical parity across protected groups defined by sensitive attributes $S$, targeting conditions such as *Demographic Parity* (DP) [1], *Equal Opportunity* (EOP) [10], or *Equalized Odds* (EOD), which enforces equality of both true positive and false positive rates [10]. Evaluation metrics typically include disparities in these statistical rates across groups and measures like *Worst-Group Accuracy* (WGA) [31]. In contrast, individual fairness requires that the model generates similar outputs for individuals who are similar with respect to a task-relevant distance metric $d(x, x')$ [5]. Assessing individual fairness often involves imposing Lipschitz constraints or testing consistency in sampled pairs, but defining a proper similarity metric $d$ remains a key challenge.

Given the inherent trade-offs between different fairness definitions [20] and the lack of a universally optimal notion, **FairNet** avoids directly optimizing for any single fairness criterion. Instead, it identifies performance-vulnerable instances (often associated with minority groups) and dynamically applies a *Contrastive Conditional LoRA* correction module to refine their internal representations and predictions. The objective of FairNet is to enhance performance for these disadvantaged subgroups, thereby indirectly improving metrics such as WGA and reducing EOD gaps, without enforcing a potentially abstract fairness constraint that may not align with the specific application context.

## 2.2 Fairness Intervention Strategies and Performance-Fairness Trade-off

Bias mitigation techniques are commonly categorized by their intervention stage within the machine learning pipeline: pre-processing (e.g., data rebalancing) [3], in-processing (e.g., fairness-aware regularization or adversarial training) [36], and post-processing (e.g., output adjustment) [10, 27]. Despite these methods targeting different stages of the ML pipeline, a pervasive challenge is the well-documented *performance-fairness trade-off* [7], where improvements in fairness are often accompanied by a decline in predictive performance. This trade-off is especially pronounced for static, global interventions that apply uniform corrections, failing to capture the *instance-level heterogeneity* of bias and risking both over- and under-correction.

**FairNet** introduces a novel *dynamic, instance-level intervention* strategy. As an in-processing method, it employs a lightweight bias detector that assesses instance representations during inference, selectively activating parameter-efficient Low-Rank Adaptation (LoRA) modules *only* for instances identified as potentially biased. This targeted correction at the *representation level* minimizes disruption to unbiased instances. By focusing corrections where they are most needed, FairNet enhances fairness—particularly for underperforming subgroups (e.g., improving WGA)—while largely preserving overall model performance, thus effectively mitigating the conventional trade-off.

## 2.3 Fairness under Varying Sensitive Attribute Constraints

Fairness algorithms vary in their reliance on sensitive attribute labels ($S$, e.g., race, gender). Many pre- and in-processing methods, such as GroupDRO [31], require **complete $S$ labels** to compute group statistics or implement group-aware regularization. However, full access to $S$ is often impractical due to privacy concerns, cost, or data scarcity. Other approaches attempt to function **without any $S$ labels** [11], relying on uncertainty or proxy attributes, though this can limit correction precision. A third line of work explores the use of **partial $S$ label information** [22]; for instance, JTT [22] leverages a small labeled validation set to guide the upweighting of challenging samples. Despite these advances, no existing method provides a unified solution that seamlessly accommodates *all* sensitive attribute scenarios—fully labeled, partially labeled, and unlabeled.

**FairNet** is designed with *inherent flexibility* to bridge this gap. Its internal bias detector can be trained under full $S$ supervision, adapt to partially labeled data by utilizing the available subset, or switch to unsupervised strategies (e.g., representation-based outlier detection) when $S$ is absent. This *versatility* allows FairNet to generalize beyond rigidly label-dependent methods, thereby enhancing its deployability in real-world scenarios.

# 3 Method

We introduce FairNet, a framework designed to dynamically correct fairness biases at the instance level directly within a pre-trained model $f_\theta(\cdot)$. FairNet aims to enhance fairness, particularly for minority subgroups, by selectively adjusting internal representations during inference, while minimizing the impact on overall task performance.

## 3.1 Framework Overview and Inference Process

FairNet enhances a base model $f_\theta$ by integrating two main components: Bias Detection modules ($D_\phi^{(l)}$) positioned after intermediate layers $l$, and Conditional LoRA modules ($L_{\text{cond\_lora}}^{(j)}$) associated with layers $j$. The detectors $D_\phi^{(l)}$, parameterized by $\phi$, monitor intermediate representations $h^{(l)}$ to identify samples that may belong to a minority sensitive group ($s = 1$), and output a risk score $p_s^{(l)}$.

The LoRA modules, parameterized by low-rank matrices $A_j \in \mathbb{R}^{r \times k}$ and $B_j \in \mathbb{R}^{d \times r}$ (with rank $r \ll \min(d, k)$), provide targeted adjustments $\Delta W_j = B_j A_j$ to the model's weights $W_j$. The core idea is that a high-risk score detected ($p_s^{(l)} > \tau$) triggers the activation of relevant LoRA modules during inference, applying corrective adjustments only when needed. The overall architecture is depicted in Figure 1. We summarize the notation in Supplementary Table 4.

The **inference process** for a given input sample $x$ proceeds as follows : First, necessary intermediate representations $h^{(l)}(x)$ are computed using the base model $f_\theta$. Second, all bias detectors $D_\phi^{(l)}$ evaluate these representations to produce risk scores $p_s^{(l)}(x)$. Third, based on these scores and a predefined threshold $\tau$, a set of LoRA modules is identified for activation. Finally, a forward pass is performed using the FairNet-enhanced model $f_{\text{FairNet}}$, where only the weights $W_j$ corresponding to the activated LoRA modules are adjusted ($W_j + \mathbb{I}(p_s^{(l)}(x) > \tau) \cdot (B_j A_j)$), to produce the final prediction $y_{\text{pred}}$.

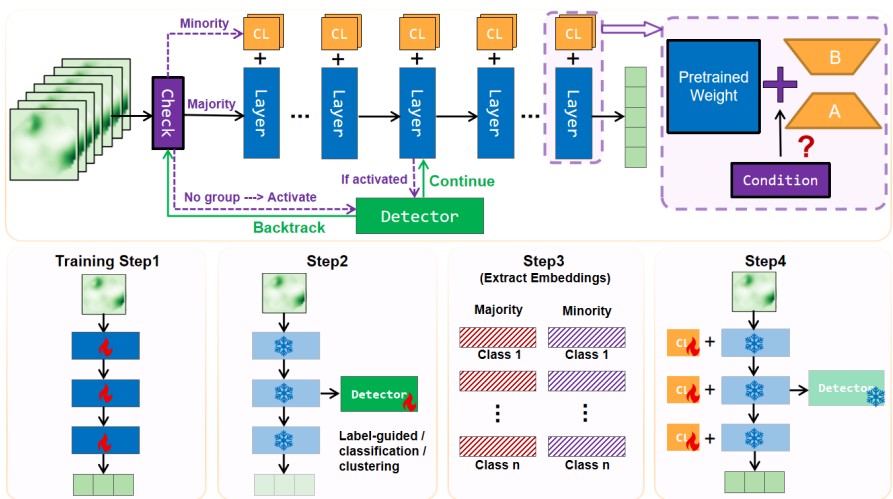

Figure 1: FairNet framework and conceptual training steps. (Top) The overall architecture featuring Bias Detectors and Contrastive Conditional LoRA (CL) modules. (Bottom) A four-step conceptual training process: (1) Base model preparation, (2) Bias Detector training, (3) Contrastive pair embedding preparation, and (4) Conditionally training LoRA modules with Contrastive Loss based on detector signals.

## 3.2 Bias Detection Module

The Bias Detector $D_\phi^{(l)}$ acts as an internal monitor at layer $l$. It takes the intermediate representation $h^{(l)}(x)$ as input and outputs a score $p_s^{(l)}(x) \in [0, 1]$ indicating the likelihood of $x$ belonging to the minority group ($s = 1$). Detectors are designed as lightweight networks (e.g., MLPs possibly preceded by attention pooling [13]) to minimize computational overhead.

Detectors are trained using a standard supervised loss, typically binary cross-entropy (BCE), on the subset of training data $P_{\text{labeled}}$ where sensitive attribute labels $s$ are available:

$$L_{\text{detector}}^{(l)} = \mathbb{E}_{(x,s) \sim P_{\text{labeled}}}[\ell_{\text{std}}(D_\phi^{(l)}(h^{(l)}(x)), s)] \tag{1}$$

This enables FairNet to operate under **partial information settings**. To counteract potential class imbalance within $P_{\text{labeled}}$ and ensure high sensitivity (TPR) to minority group samples, techniques like weighted loss or worst-group loss optimization strategies [31] can be applied during detector training. For handling **multiple sensitive attributes** $s_i$ (e.g., race $s_1$, gender $s_2$), separate detectors $D_{\phi_i}^{(l)}$ are trained for each attribute using the corresponding available labels. Each detector $D_{\phi_i}^{(l)}$ independently assesses the risk related to attribute $s_i$. Alternatively, a single detector can be employed using multi-label prediction.

### 3.3 Contrastive Conditional LoRA Module

The Conditional LoRA module $L^{(j)}_{\text{cond\_lora}}$ provides the mechanism for the correction of the targeted representation at the layer $j$. It uses the standard LoRA parameterization [13] but with two key distinctions: conditional activation (based on detector signals) and a unique training objective.

The key innovation is that the LoRA parameters $A_j, B_j$ are trained **exclusively** via a novel **contrastive loss**, $L^{(j)}_{\text{contrastive}}$. This loss directly targets the fairness goal by aiming to **minimize intra-class, inter-group representation gaps**. Specifically, it encourages the LoRA-adjusted representations $z^{(j)}(x) = \text{BaseOutput}_j(x) + \mathbb{I}(\text{trigger}^{(j)}) \cdot (B_j A_j)(\text{input}_j)$ to be similar for samples $(x_{\text{min}}, x_{\text{maj}})$ that share the same true class label $y$ but belong to different sensitive groups ($s = 1$ vs. $s = 0$). We employ a triplet loss formulation:

$$L^{(j)}_{\text{contrastive}}(x_a, x_p, x_n) = [\underbrace{D(z^{(j)}(x_a), z^{(j)}(x_p))}_{\text{Intra-class, Inter-group dist.}} - \underbrace{D(z^{(j)}(x_a), z^{(j)}(x_n))}_{\text{Anchor-Negative dist.}} + \text{margin}]_+ \qquad (2)$$

where $x_a$ is a minority anchor (label $y_a$, $s = 1$), $x_p$ is a majority positive (label $y_p = y_a$, $s = 0$), $x_n$ is a negative sample, $D(\cdot, \cdot)$ is a distance metric (e.g., squared Euclidean distance), $[\cdot]_+ = \max(0, \cdot)$, and margin $> 0$ is a hyperparameter. The selection of $x_p$ (same class, different group) is crucial for achieving the targeted representation alignment. For **multiple sensitive attributes** $s_i$, separate LoRA modules $L^{(j)}_{\text{cond\_lora\_i}}$ (with parameters $A_{j,i}, B_{j,i}$) are introduced. Each module $L^{(j)}_{\text{cond\_lora\_i}}$ is trained using an attribute-specific contrastive loss $L^{(j)}_{\text{contrastive\_i}}$ which aims to align representations across groups defined by attribute $s_i$, triggered by the corresponding detector $D^{(l)}_{\phi_i}$.

### 3.4 Model Training

The training process optimizes the base model parameters $\theta$ (optional fine-tuning), detector parameters $\phi$, and LoRA parameters $A_j, B_j$. This optimization proceeds through a multi-stage approach aimed at minimizing a composite objective function:

$$L_{\text{total}} = L_{\text{task}}(f_{\text{FairNet}}(x), y) + \lambda_D \sum_l L^{(l)}_{\text{detector}} + \lambda_C \sum_j I(\text{trigger}^{(j)}) L^{(j)}_{\text{contrastive}} \qquad (3)$$

Here, $L_{\text{task}}$ (e.g., cross-entropy) ensures the model maintains task accuracy. $L^{(l)}_{\text{detector}}$ trains the bias detectors on available labeled data $P_{\text{labeled}}$. The crucial term $L^{(j)}_{\text{contrastive}}$ drives the fairness correction via representation alignment, and its contribution is gated by the indicator $I(\text{trigger}^{(j)})$, which is 1 only if the contrastive loss of the $j$-th LoRA module was activated by a downstream detector for the current batch anchors, and 0 otherwise. $\lambda_D, \lambda_C \geq 0$ are hyperparameters that balance these objectives.

The conceptual training encompasses four sequential stages as depicted in Figure 1 (Bottom): (1) Base Model Preparation, which involves initializing or establishing the foundational model, potentially leveraging pre-trained parameters; (2) Bias Detector Training, entailing the development and training of specialized detector modules responsible for identifying inputs likely belonging to predefined (minority) subgroups based on intermediate representations; (3) Contrastive Pair Embedding Preparation, which consists of generating and processing embeddings from input samples to construct the necessary positive and negative pairs (or triplets) required for the contrastive learning objective; and (4) Conditional LoRA Module Training, which involves the fine-tuning of LoRA modules using a contrastive loss formulation, where the application of LoRA-based adjustments is dynamically triggered by the signals originating from the trained bias detectors.

## 4 Theoretical Analysis

In this section, we provide a theoretical analysis of FairNet. We first analyze how the proposed contrastive loss fosters representation fairness. Then, we derive conditions under which FairNet can provably improve fairness—specifically worst-group performance—without sacrificing, and potentially even enhancing, overall model accuracy. This analysis distinguishes our method from conventional approaches that are often subject to a performance-fairness trade-off. The detailed mathematical notations for all theorems are provided in the Supplementary Material A.

## 4.1 Preliminaries

We consider data drawn from a joint distribution $P(X, Y, S)$, where $X \in \mathcal{X}$ are inputs, $Y \in \mathcal{Y}$ are task labels, and $S \in \{0, 1\}$ is a binary sensitive attribute. We denote the majority group ($S = 0$) as $G_1$ and the minority group ($S = 1$) as $G_2$, with $P(S = 1) = p$. Let $M = f_\theta$ be the base pre-trained model and $M_{\text{FairNet}}$ be the model enhanced with FairNet's bias detectors $D_\phi^{(l)}$ and conditional LoRA modules $L_{\text{cond\_lora}}^{(j)}$ (parameters $A_j, B_j$). Let $h^{(l)}(x)$ be the representation at layer $l$ in the base model, and $z^{(j)}(x)$ be the representation at layer $j$ potentially adjusted by $L_{\text{cond\_lora}}^{(j)}$.

We evaluate performance using group-conditional accuracy $P(M, G) = P(\hat{Y} = Y | G)$ and overall accuracy $P(M) = (1 - p)P(M, G_1) + pP(M, G_2)$, where $\hat{Y} = M(X)$. Fairness is assessed using metrics like WGA, $WGA(M) = \min_{G \in \{G_1, G_2\}} P(M, G)$, and conditional fairness metrics like EOD. The bias detector $D_\phi^{(l)}$ is characterized by its True Positive Rate ($TPR_D = P(D_\phi^{(l)}(h^{(l)}) > \tau | S = 1)$) and False Positive Rate ($FPR_D = P(D_\phi^{(l)}(h^{(l)}) > \tau | S = 0)$), where $\tau$ is the activation threshold. The detailed derivations are provided in the Supplementary Material B.

## 4.2 Contrastive Loss and Representation Fairness

The core mechanism for improving fairness on FairNet is the custom contrastive loss $L_{\text{contrastive}}^{(j)}$ (Eq. 2) used to train the conditional LoRA modules. This loss targets **intra-class, inter-group representation alignment**. By minimizing the distance $D(z^{(j)}(x_a), z^{(j)}(x_p))$ between samples $x_a$ (minority, label $y$) and $x_p$ (majority, label $y$), the training process encourages the adjusted representations $z^{(j)}$ to become invariant to the sensitive attribute $s$, conditional on the true class label $y$.

Successful minimization of $L_{\text{contrastive}}^{(j)}$ implies that for a given class $y$, the distribution of representations $P(z^{(j)} | Y = y, S = 0)$ approaches $P(z^{(j)} | Y = y, S = 1)$. Theoretical results in fair representation learning suggest that, firstly, by enabling minority group representations to learn from those of the (often better-performing) majority group, the performance of the minority group is typically enhanced (which in turn can improve WGA), and secondly, if representations are conditionally independent of the sensitive attribute given the true label, then any classifier relying solely on these representations will satisfy conditional fairness criteria like EOp and EOD. Specifically, reducing the discrepancy between $P(z^{(j)} | Y = y, S = 0)$ and $P(z^{(j)} | Y = y, S = 1)$ is expected to bound the downstream EOD. Thus, FairNet's contrastive loss directly targets the representational disparities that underpin conditional fairness violations.

## 4.3 Performance Preservation Analysis

A critical aspect of FairNet is its ability to enhance fairness without degrading overall performance $P(M)$. We analyze the change in overall performance $\Delta P = P(M_{\text{FairNet}}) - P(M)$. Let $P(M_{\text{LoRA}}, G)$ denote the hypothetical accuracy if the contrastively trained LoRA modules were unconditionally applied to all samples from the group $G$. The actual performance of FairNet on each group depends on the detector's rates ($TPR_D, FPR_D$):

$$P(M_{\text{FairNet}}, G_1) = (1 - FPR_D) \cdot P(M, G_1) + FPR_D \cdot P(M_{\text{LoRA}}, G_1) \tag{4}$$

$$P(M_{\text{FairNet}}, G_2) = TPR_D \cdot P(M_{\text{LoRA}}, G_2) + (1 - TPR_D) \cdot P(M, G_2) \tag{5}$$

These follow from partitioning each group based on whether the detector triggers the LoRA correction (correctly for $G_2$ with probability $TPR_D$, incorrectly for $G_1$ with probability $FPR_D$).

The overall performance change is the weighted average of group performance changes:

$$\Delta P = (1 - p)[P(M_{\text{FairNet}}, G_1) - P(M, G_1)] + p[P(M_{\text{FairNet}}, G_2) - P(M, G_2)] \tag{6}$$

$$= (1 - p)FPR_D[P(M_{\text{LoRA}}, G_1) - P(M, G_1)] + pTPR_D[P(M_{\text{LoRA}}, G_2) - P(M, G_2)] \tag{7}$$

To ensure non-decreasing performance ($\Delta P \geq 0$), assuming $P(M_{\text{LoRA}}, G_2) - P(M, G_2) > 0$ (LoRA improves minority performance) and $FPR_D > 0$, rearranging Eq. 7 yields the condition:

$$\frac{TPR_D}{FPR_D} \geq \frac{1 - p}{p} \cdot \frac{P(M, G_1) - P(M_{\text{LoRA}}, G_1)}{P(M_{\text{LoRA}}, G_2) - P(M, G_2)} \tag{8}$$

This condition relates the detector's quality (ratio $TPR_D/FPR_D$) to the relative population sizes $((1 - p)/p)$ and the differential impact of the LoRA correction on the two groups.

**Why FairNet Satisfies the Condition:** FairNet's design, particularly its contrastive LoRA training objective, makes it highly likely to satisfy Condition 8:

1. **Large Positive Denominator:** $P(M_{\text{LoRA}}, G_2) - P(M, G_2)$ is expected to be significantly positive. The base model $M$ often performs poorly in the minority group $G_2$. The contrastive loss $L_{\text{contrastive}}$ explicitly trains the LoRA modules to improve $G_2$'s representations by aligning them with $G_1$'s (presumably better) representations within each class. This targeted improvement should substantially increase $P(M_{\text{LoRA}}, G_2)$ over $P(M, G_2)$.

2. **Small or Negative Numerator:** $P(M, G_1) - P(M_{\text{LoRA}}, G_1)$ is expected to be small or even negative. Since the contrastive alignment target is the majority group's representation space, applying the resulting LoRA correction (which only happens for $G_1$ samples due to detector false positives) should ideally have minimal negative impact on $G_1$'s performance. The correction might slightly perturb $G_1$'s representations, potentially causing a small performance drop ($P(M_{\text{LoRA}}, G_1) < P(M, G_1)$), but this effect is anticipated to be much smaller than the gain for $G_2$. It is even possible that the alignment process slightly regularizes or improves the robustness of $G_1$'s representations, leading to $P(M_{\text{LoRA}}, G_1) \geq P(M, G_1)$, making the numerator non-positive.

3. **Achievable Condition:** Consequently, the ratio $\frac{P(M,G_1)-P(M_{\text{LoRA}},G_1)}{P(M_{\text{LoRA}},G_2)-P(M,G_2)}$ is expected to be a small positive value, zero, or negative. As long as the bias detector has reasonable discriminative ability, the inequality in Condition 8 is likely to hold.

**Contrast with Other Architectures:** Many existing bias mitigation strategies lack an explicit internal detection mechanism, rendering them incapable of exploiting the theoretical performance preservation condition (Condition 8), which fundamentally hinges on the detector's efficacy.

Conventional bias-mitigation methods uniformly apply global or static interventions and thus cannot escape the inherent performance–fairness trade-off. Pre-processing techniques rebalance data via resampling or reweighting—thereby perturbing the original distribution and impairing generalization—while in-processing approaches embed fairness constraints into training, which often conflict with the primary optimization objective and hinder the learning of optimal decision boundaries. Post-processing adjustments rectify outputs to meet fairness metrics but leave biased internal representations intact and may degrade predictive accuracy or impair calibration.

FairNet's architecture—explicitly integrating an internal bias detector with a conditional correction module—is expressly designed to circumvent this limitation.

# 5   Experiments

We empirically assess FairNet by (i) benchmarking its accuracy–fairness trade-off against strong SOTA baselines and (ii) running targeted ablations that isolate the impact of its key components. Additional experiments and analyses appear in the Supplemental Material D.

## 5.1   Experimental Setup

We evaluate FairNet on three diverse datasets: CelebA [24], MultiNLI [34], and HateXplain [26], which represent different modalities and bias types. For CelebA, we predict the "Male" attribute while accounting for "Blond Hair" as a sensitive attribute, revealing imbalances across male and female images with blond hair. In MultiNLI, we predict entailment relations with a focus on negation as a sensitive attribute, uncovering linguistic biases. HateXplain helps assess overlapping biases related to gender and race, focusing on hate speech prediction. We fine-tune ViT for vision tasks and BERT for language tasks; full experimental details are provided in the Supplementary Material C.

We evaluate three versions of FairNet based on the availability of sensitive attribute labels.

- **FairNet-Full** assumes complete sensitive attribute labels ($S$) are available for both training and validation; its bias detector ($D_\phi^{(l)}$) directly uses these ground-truth sensitive labels.

- **FairNet-Partial** is designed for scenarios where the training set contains only partial sensitive attribute labels (e.g., $k\%$ of samples are labeled with $S$), and the validation set has no $S$ labels. In this setting, the bias detector is trained on the labeled subset of the training data, and the contrastive LoRA module utilizes these available labels.

- **FairNet-Unlabeled** addresses cases where no sensitive attribute labels are available in either training or validation. Here, the bias detector ($D_\phi^{(l)}$) employs unsupervised methods (outlier detection on $h^{(l)}(x)$) to generate pseudo-sensitive attribute labels ($\hat{s}$), which are then used by the $L_{\text{cond\_lora}}^{(j)}$ to form corrective pairs.

## 5.2 Comparison with Existing Methods

We benchmark FairNet against a representative spectrum of bias-mitigation baselines that span the three canonical intervention stages— pre-processing, in-processing, and post-processing. We further stratify methods based on the degree of sensitive-attribute availability into three scenarios: (1) no access to sensitive labels during both training and testing; (2) access to sensitive labels during training but not at test time; and (3) access to sensitive labels at both training and testing. To assess performance and fairness, we primarily report overall task accuracy (ACC), WGA to evaluate the performance on the most disadvantaged group, and the EOD to measure inter-group disparities in true positive and false positive rates across two distinct datasets, CelebA and MultiNLI.

Table 1: Performance comparison across different datasets and methods.

| Method | Group Info Train / Test | CelebA(%) | | | MultiNLI(%) | | |
|---|---|---|---|---|---|---|---|
| | | ACC↑ | WGA↑ | EOD↓ | ACC↑ | WGA↑ | EOD↓ |
| ERM | × / × | 95.8 | 77.9 | 10.6 | 82.6 | 67.3 | 12.5 |
| Lu et al.[25] | × / × | 95.4 | 81.4 | 8.3 | 82.0 | 72.8 | 8.5 |
| D3M[15] | × / × | 95.2 | 82.0 | 8.1 | 81.0 | 72.8 | 8.3 |
| **FairNet-Unlabel** | × / × | 95.8 | 82.3 | 7.3 | 82.5 | 73.1 | 8.1 |
| GroupDRO | ✓ / × | 94.0 | 87.4 | 4.7 | 80.8 | 78.2 | 5.5 |
| DFR[19] | partial / × | 94.3 | 86.0 | 7.7 | 81.2 | 74.1 | 6.7 |
| Sebra[18] | partial / × | 94.8 | 85.2 | 8.1 | 81.5 | 74.2 | 6.5 |
| **FairNet-Partial** | partial / × | **95.9** | 86.5 | 5.6 | **82.6** | 76.5 | 6.2 |
| Eq.Odds[10] | ✓ / ✓ | 95.0 | 83.2 | 7.2 | 81.3 | 75.3 | 6.3 |
| GSTAR[16] | ✓ / ✓ | 94.2 | 85.4 | 6.6 | 80.8 | 76.6 | 6.2 |
| **FairNet-Full** | ✓ / ✓ | **95.9** | **88.2** | **3.8** | **82.6** | **78.5** | **4.7** |

[*] **Bold** indicates the global best performance; Underlined indicates the best in each category.

**No Access to Sensitive Attributes:** In the absence of sensitive attributes, **FairNet-Unlabel** demonstrates strong performance. On **CelebA**, it achieves an ACC of $95.8\%$, matching ERM, while attaining the highest WGA ($82.3\%$) and the lowest EOD ($7.3\%$). On **MultiNLI**, FairNet-Unlabel again achieves competitive ACC ($82.5\%$, matching ERM) and secures the best WGA ($73.1\%$) and EOD ($8.1\%$) among methods in this category. These results highlight FairNet's capacity to improve fairness in the absence of explicit group supervision, without compromising overall performance.

**Partial Access to Sensitive Attributes:** When sensitive attributes are partially available during training, **FairNet-Partial** yields the highest ACC on both **CelebA** ($95.9\%$) and **MultiNLI** ($82.6\%$). On **CelebA**, its WGA ($86.5\%$) and EOD ($5.6\%$) are competitive, ranking second to GroupDRO (WGA $87.4\%$, EOD $4.7\%$) but notably improving over DFR and Sebra. On **MultiNLI**, FairNet-Partial achieves a WGA of $76.5\%$ and an EOD of $6.2\%$, again demonstrating a strong balance of accuracy and fairness, outperforming DFR and Sebra in these aspects.

**Full Access to Sensitive Attributes:** Given full access to sensitive labels during both training and testing, **FairNet-Full** consistently outperforms all other methods across all metrics. On **CelebA**, it achieves state-of-the-art results with ACC $95.9\%$, WGA $88.2\%$, and EOD $3.8\%$. Similarly, on **MultiNLI**, FairNet-Full leads with ACC $82.6\%$, WGA $78.5\%$, and EOD $4.7\%$. These results highlight FairNet's efficacy when complete group information is available.

**Overall Comparison:** Across all evaluated scenarios, FairNet variants demonstrate a compelling balance between overall accuracy and fairness metrics. FairNet effectively adapts to varying levels of sensitive attribute availability, consistently improving WGA and reducing EOD relative to baselines, often while maintaining or exceeding their ACC. The performance on fairness metrics generally improves with increased access to sensitive information, showcasing the robustness and versatility of the FairNet framework.

### 5.3 Efficacy in Mitigating Intersectional Bias

Beyond single-axis biases, real-world applications demand fairness across multiple, often intersecting, sensitive attributes such as race and gender. To evaluate FairNet's capabilities in such complex scenarios, we conducted experiments on the HateXplain dataset, which contains multifaceted biases. Using FairNet-Partial, we applied a progressive debiasing strategy: first targeting racial bias (related to the "African American" demographic) and subsequently targeting gender bias (related to the "Female" demographic).

Table 2: Performance and fairness during progressive debiasing on the HateXplain dataset.

| Metric | DistilBERT-base | | | BERT-base | | |
|---|---|---|---|---|---|---|
| | ERM | FairNet Afr. | FairNet Fe. | ERM | FairNet Afr. | FairNet Fe. |
| DP (R)$\downarrow$ | $38.2 \pm 1.4$ | $33.7 \pm 1.4$ | $32.8 \pm 1.1$ | $27.1 \pm 0.9$ | $14.0 \pm 1.0$ | $12.4 \pm 0.7$ |
| EOp (R)$\downarrow$ | $14.9 \pm 1.1$ | $14.2 \pm 1.0$ | $13.1 \pm 1.0$ | $13.0 \pm 0.8$ | $8.4 \pm 1.1$ | $7.2 \pm 1.0$ |
| **EOD (R)**$\downarrow$ | $26.5 \pm 0.7$ | $\underline{24.4 \pm 0.6}$ | $23.0 \pm 0.6$ | $20.1 \pm 0.4$ | $\underline{11.2 \pm 0.6}$ | $9.8 \pm 0.5$ |
| DP (G)$\downarrow$ | $7.4 \pm 1.3$ | $7.6 \pm 1.1$ | $12.9 \pm 2.2$ | $7.6 \pm 1.5$ | $8.5 \pm 1.4$ | $7.6 \pm 1.0$ |
| EOp (G)$\downarrow$ | $13.0 \pm 0.5$ | $13.0 \pm 0.5$ | $2.0 \pm 2.1$ | $18.2 \pm 0.8$ | $16.7 \pm 0.4$ | $8.8 \pm 1.4$ |
| **EOD (G)**$\downarrow$ | $11.3 \pm 1.1$ | $\underline{11.2 \pm 0.7}$ | $7.4 \pm 0.6$ | $12.9 \pm 1.1$ | $\underline{12.6 \pm 0.9}$ | $8.2 \pm 0.4$ |
| **ACC**$\uparrow$ | $79.5 \pm 0.2$ | $\underline{79.6 \pm 0.2}$ | $79.7 \pm 0.3$ | $79.8 \pm 0.3$ | $79.6 \pm 0.5$ | $\underline{79.7 \pm 0.4}$ |

[*] Bold values indicate the best performance in each category, while underlined values represent the second-best results. "R" refers to Race (African American vs. Other), and "G" refers to Gender (Female vs. Male).

The results, presented in Table 2, demonstrate FairNet's capacity for effective, progressive debiasing. Applying the correction for the "African American" group (**FairNet Afr.**) successfully reduces racial bias across both models. For DistilBERT-base, the EOD (R) improves from 26.5% to 24.4%, while for BERT-base, the reduction is even more pronounced, from 20.1% to 11.2%.

Crucially, the subsequent application of a fairness module for the "Female" group (**FairNet Fe.**) not only preserves or enhances the initial gains in racial fairness but also delivers substantial improvements in gender fairness. This progressive approach further lowers EOD (R) to 23.0% for DistilBERT and 9.8% for BERT-base. Simultaneously, it dramatically reduces gender-based disparity, with EOD (G) dropping from 11.3% to 7.4% on DistilBERT, and from 12.9% to 8.2% on BERT-base.

This multi-faceted bias mitigation is achieved without compromising task performance. The overall accuracy is consistently maintained or slightly improved across all interventions for both models. These findings validate the modularity and adaptability of the FairNet architecture, showcasing its ability to address complex, multi-attribute fairness challenges in a targeted manner while upholding overall model performance. This is a vital characteristic for real-world deployments where fairness considerations often span several demographic dimensions.

### 5.4 Ablation Studies

To quantify the contribution of **FairNet**'s core components—the bias detector and the contrastive loss—we conduct ablation experiments on the CelebA and MultiNLI datasets. We evaluate the impact of removing each of these components individually, as well as removing both simultaneously. All experiments are conducted under a *partial* sensitive-attribute setting, in which both the full-attribute-label and no-sensitive-attribute-label scenarios are treated as special cases of partial observation. The primary controlled ablations focus on:

1. **Detector Ablation:** We disable the bias detector $D_\phi^{(l)}$, so that conditional LoRA modules remain active on every input. This variant measures how selective gating via the detector preserves accuracy on non-biased instances while still applying corrections where required.

2. **Contrastive-Loss Ablation:** We replace our contrastive loss $L_{\text{contrastive}}^{(j)}$ with binary cross entropy loss on flagged instances. This experiment evaluates the role of the contrastive objective in closing intra-class representation gaps and boosting minority-group performance.

Table 3: Ablation study results of FairNet on CelebA and MultiNLI datasets.

| Method | CelebA | | | MultiNLI | | |
|---|---|---|---|---|---|---|
| | ACC (%) | WGA (%) | EOD (%) | ACC (%) | WGA (%) | EOD (%) |
| FairNet-Partial | **95.9** | 86.5 | 5.6 | **82.6** | 76.5 | 6.2 |
| w/o detector | 94.1 | **86.7** | **5.3** | 81.0 | **77.2** | **5.7** |
| w/o contrastive loss | 95.8 | 81.2 | 8.5 | 82.5 | 70.1 | 9.2 |
| w/o both | 94.3 | 82.3 | 7.8 | 81.3 | 71.8 | 8.9 |
| ERM | 95.8 | 77.9 | 10.6 | **82.6** | 67.3 | 12.5 |

[*]**Bold** indicates the best; Underlined indicates second-best.

The results of our ablation studies (Table 3) reveal the consistent contributions of **FairNet**'s components across both datasets. Disabling the bias detector ("w/o detector") leads to a notable ACC drop on both CelebA (from 95.9% to 94.1%) and MultiNLI (from 82.6% to 81.0%), even as fairness metrics like WGA and EOD marginally improve. This highlights the detector's key role in maintaining high overall accuracy by applying corrections selectively. Conversely, removing the contrastive loss ("w/o contrastive loss") substantially degrades fairness on both datasets with only a minor change in ACC. On CelebA, WGA falls from 86.5% to 81.2%, and on MultiNLI it drops from 76.5% to 70.1%. This confirms the critical function of $L_{\text{contrastive}}^{(j)}$ in improving minority group representations. The "w/o both" variant, lacking both core components, shows fairness performance superior to the ERM baseline but significantly lower than the full **FairNet** model. These findings affirm the synergistic importance of the bias detector for preserving accuracy and the contrastive loss for enhancing fairness.

# 6 Conclusion

We propose **FairNet**, a dynamic, instance-conditioned framework that reconciles fairness and accuracy by pairing lightweight bias detectors with contrastively trained conditional LoRA adapters. Unlike global debiasing schemes that impose uniform changes on every example, FairNet activates corrective updates only for samples predicted to be vulnerable, thereby preserving performance for the majority group while narrowing critical error gaps for protected subgroups.

Theoretical analysis shows that, whenever the detector achieves a sufficiently high $\text{TPR}/\text{FPR}$ ratio, FairNet provably improves worst-group performance without diminishing—and often slightly improving—overall accuracy, thus breaking the long-assumed performance–fairness trade-off. Extensive experiments on vision and language benchmarks and datasets confirm these guarantees under three practical settings: fully labeled, partially labeled, and unlabeled sensitive attributes. In nearly all cases, FairNet delivers the highest WGA and the lowest EOD while matching or surpassing strong ERM baselines in overall accuracy. Ablation studies further reveal that both selective activation and the contrastive objective are indispensable to these gains.

FairNet's modular, instance-conditioned design—by successfully decoupling fairness interventions from overall performance impacts—represents a significant advancement. Backed by both theoretical guarantees and extensive empirical validation, it offers a practical pathway towards deploying AI systems that are both high-performing and equitable, effectively moving beyond traditional compromises. Future work will address complex intersectional biases and integrate advanced unsupervised detection methods, further broadening FairNet's impact on trustworthy AI.

## Acknowledgements

This work is supported by the Tsinghua-Toyota Joint Research Fund, the National Natural Science Foundation of China (62273197), and the Beijing Natural Science Foundation (L233027).

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

# Supplementary Material

## A    Notation

We summarize the notation used in the main paper in Supplementary Table 4.

## B    Theorem Proofs and Derivations

This section provides detailed derivations and proofs for the theoretical claims made in Section 4 of the main paper.

### B.1    Contrastive Loss and Representation Fairness (Main Paper Section 4.2)

The core idea behind FairNet's contrastive loss $L_{\text{contrastive}}^{(j)}$ (Equation 2 in the main paper) is to align the representations of samples from the same class but different sensitive groups. The triplet loss formulation is:

$$L_{\text{contrastive}}^{(j)}(x_a, x_p, x_n) = [D(z^{(j)}(x_a), z^{(j)}(x_p)) - D(z^{(j)}(x_a), z^{(j)}(x_n)) + \text{margin}]_+$$

where $x_a$ is an anchor from the minority group ($S = 1$) with label $Y = y$, $x_p$ is a positive sample from the majority group ($S = 0$) with the same label $Y = y$, and $x_n$ is a negative sample from any group with a different label $Y \neq y$. The LoRA-adjusted representation is $z^{(j)}(x)$.

Minimizing this loss encourages $D(z^{(j)}(x_a), z^{(j)}(x_p))$ to be small, ideally smaller than $D(z^{(j)}(x_a), z^{(j)}(x_n))$ by at least the margin. This means that after the LoRA adjustment (which is primarily triggered for minority group samples or samples predicted as such by the detector), the representation $z^{(j)}(x_a)$ for a minority sample becomes closer to the representation $z^{(j)}(x_p)$ of a majority sample of the same class. If this minimization is successful across many such triplets, the distribution of adjusted representations for the minority group, $P(z^{(j)}|Y = y, S = 1)$, will be pushed towards the distribution of representations for the majority group, $P(z^{(j)}|Y = y, S = 0)$, for each class $y$.

**Implications for Worst-Group Accuracy (WGA):** Models often underperform for minority groups due to data scarcity or skewed data distributions, leading to suboptimal representations for these groups. By aligning minority group representations ($S = 1$) with those of the majority group ($S = 0$) within the same class $Y = y$, FairNet effectively allows the minority group to leverage the potentially better-learned representational characteristics of the majority group. If the majority group's representations are more discriminative for the downstream task, this alignment can lead to improved classification accuracy for the minority group. Since WGA is defined as $WGA(M) = \min_{G \in \{G_1, G_2\}} P(M, G)$, an improvement in the minority group's accuracy ($P(M, G_2)$) directly contributes to an increase in WGA, assuming the minority group is the worst-performing one.

**Implications for Equalized Odds Difference ( EOD):** Equalized Odds requires that the prediction $\hat{Y}$ is independent of the sensitive attribute $S$ given the true label $Y$. This means $P(\hat{Y} = 1|Y = y, S = 0) = P(\hat{Y} = 1|Y = y, S = 1)$ for $y \in \{0, 1\}$ (for binary classification and labels). If the representations $z^{(j)}$ upon which the final classifier operates are made conditionally independent of $S$ given $Y$, i.e., $P(z^{(j)}|Y = y, S = 0) \approx P(z^{(j)}|Y = y, S = 1)$, then any downstream classifier that only uses $z^{(j)}$ will naturally tend to satisfy Equalized Odds. Achieving such representational fairness is a strong step towards satisfying fairness criteria like Equalized Odds. Reducing the discrepancy between $P(z^{(j)}|Y = y, S = 0)$ and $P(z^{(j)}|Y = y, S = 1)$ aims to reduce the statistical information about $S$ in $z^{(j)}$ that is not already captured by $Y$. This, in turn, is expected to reduce the $\Delta$EOD, which measures the disparity in true positive rates and false positive rates between groups. While a direct quantitative bound on EOD from the contrastive loss value is complex, the qualitative argument is that by making the input representations to the final classification layers more similar across groups for a given true class, the classification outcomes will also become more similar, thus reducing EOD.

**Performance Preservation Analysis (Main Paper Section 4.3)**

We derive the equations presented for the performance preservation analysis.

Table 4: Summary of Notation

| Symbol | Description |
|---|---|
| $X, x$ | Input space, an input sample |
| $Y, y$ | Task label space, a task label |
| $S, s$ | Sensitive attribute space, a sensitive attribute ($s = 1$ for minority, $s = 0$ for majority) |
| $f_\theta(\cdot)$ | Base pre-trained model with parameters $\theta$ |
| $h^{(l)}(x)$ | Intermediate representation of $x$ at layer $l$ of the base model |
| $D_\phi^{(l)}$ | Bias Detection module at layer $l$ with parameters $\phi$ |
| $p_s^{(l)}(x)$ | Risk score from $D_\phi^{(l)}$ indicating likelihood of $x$ belonging to minority group |
| $\tau$ | Activation threshold for LoRA modules based on $p_s^{(l)}(x)$ |
| $L_{\text{cond\_lora}}^{(j)}$ | Conditional LoRA module associated with layer $j$ |
| $W_j$ | Original weights of layer $j$ in the base model |
| $A_j, B_j$ | Low-rank matrices for LoRA module at layer $j$ ($A_j \in \mathbb{R}^{r \times k}$, $B_j \in \mathbb{R}^{d \times r}$) |
| $r$ | Rank of LoRA matrices ($r \ll \min(d, k)$) |
| $\Delta W_j$ | Weight adjustment from LoRA module, $\Delta W_j = B_j A_j$ |
| $f_{\text{FairNet}}$ | FairNet-enhanced model |
| $P_{\text{labeled}}$ | Subset of training data with available sensitive attribute labels |
| $\ell_{\text{std}}$ | Standard supervised loss (e.g., Binary Cross-Entropy) |
| $s_i$ | $i$-th sensitive attribute (for multiple attributes) |
| $D_{\phi_i}^{(l)}$ | Bias detector for sensitive attribute $s_i$ |
| $z^{(j)}(x)$ | LoRA-adjusted representation at layer $j$ |
| $x_a, x_p, x_n$ | Anchor, positive, and negative samples for contrastive loss |
| $D(\cdot, \cdot)$ | Distance metric for contrastive loss (e.g., squared Euclidean distance) |
| margin | Margin hyperparameter for triplet contrastive loss |
| $L_{\text{task}}$ | Task-specific loss (e.g., cross-entropy) |
| $L_{\text{detector}}^{(l)}$ | Loss for training bias detector $D_\phi^{(l)}$ (Eq. 1) |
| $L_{\text{contrastive}}^{(j)}$ | Contrastive loss for training LoRA module $L_{\text{cond\_lora}}^{(j)}$ (Eq. 2) |
| $I(\text{trigger}^{(j)})$ | Indicator function for activation of $j$-th LoRA module's contrastive loss during training |
| $L_{\text{total}}$ | Total composite loss function for FairNet training (Eq. 3) |
| $\lambda_D, \lambda_C$ | Hyperparameters balancing terms in $L_{\text{total}}$ |
| $P(X, Y, S)$ | Joint data distribution |
| $\mathcal{X}, \mathcal{Y}$ | Input and task label spaces, respectively |
| $G_1, G_2$ | Majority group ($S = 0$), Minority group ($S = 1$) |
| $p$ | Prior probability of belonging to the minority group, $P(S = 1)$ |
| $M$ | Base model $f_\theta$ |
| $M_{\text{FairNet}}$ | FairNet model |
| $P(M, G)$ | Accuracy of model $M$ on group $G$ |
| $P(M)$ | Overall accuracy of model $M$ |
| $\hat{Y}$ | Predicted label by a model |
| $\text{WGA}(M)$ | Worst-Group Accuracy of model $M$ |
| EOD | Equalized Odds |
| $TPR_D$ | True Positive Rate of the bias detector |
| $FPR_D$ | False Positive Rate of the bias detector |
| $P(M_{\text{LoRA}}, G)$ | Hypothetical accuracy if LoRA modules were unconditionally applied to group $G$ |
| $\Delta P$ | Change in overall performance $P(M_{\text{FairNet}}) - P(M)$ |
| ACC | Overall task accuracy |

**Derivation of Equations 4 and 5: Performance of FairNet on each group**

The bias detector $D_\phi^{(l)}$ produces a binary signal indicating whether a sample is suspected of being from a disadvantaged group. We define $TPR_D = P(D_\phi^{(l)} \text{ fires} \mid S = 1)$ and $FPR_D = P(D_\phi^{(l)} \text{ fires} \mid S = 0)$. The contrastive LoRA correction is conditionally activated only when the detector fires.

For group $G_1$ (majority, $S = 0$):

- With probability $FPR_D$, the detector incorrectly fires, and LoRA is applied. The accuracy in this case is $P(M_{\text{LoRA}}, G_1)$.
- With probability $(1 - FPR_D)$, the detector correctly does not fire, and the base model $M$ is used. The accuracy is $P(M, G_1)$.

So, the performance for group $G_1$ (Equation 4) is:

$$P(M_{\text{FairNet}}, G_1) = FPR_D \cdot P(M_{\text{LoRA}}, G_1) + (1 - FPR_D) \cdot P(M, G_1)$$

For group $G_2$ (minority, $S = 1$):

- With probability $TPR_D$, the detector correctly fires, and LoRA is applied. The accuracy is $P(M_{\text{LoRA}}, G_2)$.
- With probability $(1 - TPR_D)$, the detector incorrectly does not fire, and the base model $M$ is used. The accuracy is $P(M, G_2)$.

So, the performance for group $G_2$ (Equation 5) is:

$$P(M_{\text{FairNet}}, G_2) = TPR_D \cdot P(M_{\text{LoRA}}, G_2) + (1 - TPR_D) \cdot P(M, G_2)$$

**Derivation of Equation 6 and 7: Overall performance change $\Delta P$**

The overall accuracy of a model $M'$ is given by:

$$P(M') = (1 - p)P(M', G_1) + pP(M', G_2)$$

where $p = P(S = 1)$. The change in overall performance is:

$$\Delta P = P(M_{\text{FairNet}}) - P(M)$$

Substituting the expressions for $P(M_{\text{FairNet}})$ and $P(M)$:

$$\Delta P = [(1 - p)P(M_{\text{FairNet}}, G_1) + pP(M_{\text{FairNet}}, G_2)] - [(1 - p)P(M, G_1) + pP(M, G_2)]$$
$$= (1 - p)[P(M_{\text{FairNet}}, G_1) - P(M, G_1)] + p[P(M_{\text{FairNet}}, G_2) - P(M, G_2)]$$

Substitute Equations 4 and 5 into the terms above:

For the first term, representing the change in performance for group $G_1$:

$$P(M_{\text{FairNet}}, G_1) - P(M, G_1) = FPR_D \cdot P(M_{\text{LoRA}}, G_1) + (1 - FPR_D) \cdot P(M, G_1) - P(M, G_1)$$
$$= FPR_D \cdot P(M_{\text{LoRA}}, G_1) - FPR_D \cdot P(M, G_1)$$
$$= FPR_D[P(M_{\text{LoRA}}, G_1) - P(M, G_1)]$$

For the second term, representing the change in performance for group $G_2$:

$$P(M_{\text{FairNet}}, G_2) - P(M, G_2) = TPR_D \cdot P(M_{\text{LoRA}}, G_2) + (1 - TPR_D) \cdot P(M, G_2) - P(M, G_2)$$
$$= TPR_D \cdot P(M_{\text{LoRA}}, G_2) - TPR_D \cdot P(M, G_2)$$
$$= TPR_D[P(M_{\text{LoRA}}, G_2) - P(M, G_2)]$$

Substituting these back into the expression for $\Delta P$, we get Equation 7:

$$\Delta P = (1 - p)FPR_D[P(M_{\text{LoRA}}, G_1) - P(M, G_1)] + pTPR_D[P(M_{\text{LoRA}}, G_2) - P(M, G_2)]$$

**Derivation of Equation 8: Condition for $\Delta P \geq 0$**

We want to find the condition for non-decreasing performance, i.e., $\Delta P \geq 0$.

$$(1 - p)FPR_D[P(M_{\text{LoRA}}, G_1) - P(M, G_1)] + pTPR_D[P(M_{\text{LoRA}}, G_2) - P(M, G_2)] \geq 0$$

Assume $P(M_{\text{LoRA}}, G_2) - P(M, G_2) > 0$ (LoRA improves minority performance, which is the goal). Also assume $FPR_D > 0$ (the detector is not perfect and sometimes misfires on the majority group). Rearranging the terms:

$$pTPR_D[P(M_{\text{LoRA}}, G_2) - P(M, G_2)] \geq -(1 - p)FPR_D[P(M_{\text{LoRA}}, G_1) - P(M, G_1)]$$

$$pTPR_D[P(M_{\text{LoRA}}, G_2) - P(M, G_2)] \geq (1 - p)FPR_D[P(M, G_1) - P(M_{\text{LoRA}}, G_1)]$$

Dividing by $p \cdot FPR_D \cdot [P(M_{\text{LoRA}}, G_2) - P(M, G_2)]$ (this term is positive under our assumptions), we obtain Equation 8:

$$\frac{TPR_D}{FPR_D} \geq \frac{(1 - p)}{p} \frac{[P(M, G_1) - P(M_{\text{LoRA}}, G_1)]}{[P(M_{\text{LoRA}}, G_2) - P(M, G_2)]}$$

**Justification for "Why FairNet Satisfies the Condition" (Elaboration)**

The main paper (Section 4.3) provides three key arguments. We elaborate further:

1. **Large Positive Denominator:** $P(M_{\textbf{LoRA}}, G_2) - P(M, G_2)$ **is significantly positive.** The base model $M$ often shows disparate performance, with $P(M, G_2)$ being lower than $P(M, G_1)$. The contrastive LoRA modules are specifically trained to improve the representations of minority group samples ($G_2$) by aligning them with the (presumably better) representations of majority group samples ($G_1$) within the same class. This targeted representational enhancement is designed to directly address the underfitting or misrepresentation of $G_2$ samples. Therefore, $P(M_{\text{LoRA}}, G_2)$, the accuracy if LoRA were always applied to $G_2$, is expected to be substantially higher than $P(M, G_2)$. The magnitude of this improvement, $P(M_{\text{LoRA}}, G_2) - P(M, G_2)$, forms the denominator and is expected to be a positive and non-trivial value.

2. **Small or Negative Numerator:** $P(M, G_1) - P(M_{\textbf{LoRA}}, G_1)$ **is small or negative.** The LoRA correction $\Delta W_j$ is learned via the contrastive loss, which uses majority group ($G_1$) representations as targets for alignment within each class. When this correction is (incorrectly, due to $FPR_D$) applied to a $G_1$ sample, it aims to shift its representation $h^{(l)}(x)$ to $z^{(l)}(x)$ in a way that is consistent with the learned alignment. Ideally, if $G_1$ representations are already optimal or near-optimal for the base model, applying a LoRA adjustment might slightly perturb them, potentially leading to a small drop in performance, making $P(M_{\text{LoRA}}, G_1) < P(M, G_1)$, and thus $P(M, G_1) - P(M_{\text{LoRA}}, G_1)$ a small positive value. However, it is also plausible that the contrastive learning process, by emphasizing robust intra-class similarities, could act as a form of regularization. The LoRA adjustment, even when applied to $G_1$ samples, might push their representations towards a more generalizable or robust manifold shared with $G_2$ samples of the same class. This could potentially lead to no change or even a slight improvement in $G_1$ performance, i.e., $P(M_{\text{LoRA}}, G_1) \geq P(M, G_1)$. In this scenario, $P(M, G_1) - P(M_{\text{LoRA}}, G_1)$ would be zero or negative. In either case, the impact on $G_1$ is expected to be much less detrimental than the positive impact on $G_2$, as the LoRA modules are not trained to specifically alter $G_1$ representations away from their original effective state in a harmful way.

3. **Achievable Condition:** Given point 1 (denominator is significantly positive) and point 2 (numerator is small positive, zero, or negative), the fraction

$$\frac{P(M, G_1) - P(M_{\text{LoRA}}, G_1)}{P(M_{\text{LoRA}}, G_2) - P(M, G_2)}$$

will be a small number (possibly close to zero or negative). The term

$$\frac{1 - p}{p}$$

is the ratio of majority to minority group sizes, which is a constant for a given dataset.
If the numerator is negative or zero (i.e., LoRA does not hurt or helps $G_1$), then the right-hand side of Equation 8 is $\leq 0$. Since $TPR_D/FPR_D$ is always non-negative, the condition holds trivially for any reasonable detector.
If the numerator is small and positive, the condition becomes

$$\frac{TPR_D}{FPR_D} \geq \text{small\_positive\_value}.$$

This is a realistic condition to meet for a bias detector that has even moderate discriminative power (i.e., $TPR_D$ is reasonably high and $FPR_D$ is reasonably low, making $TPR_D/FPR_D$ significantly greater than 1). For instance, if the accuracy loss on $G_1$ due to LoRA is much smaller than the accuracy gain on $G_2$, then the ratio on the right side is small.

Therefore, FairNet's design, which focuses on improving $G_2$ performance while minimally affecting $G_1$ (due to targeted training of LoRA and conditional application), makes Condition 8 practically achievable. This analysis supports the claim that FairNet can improve fairness (specifically WGA by increasing $P(M, G_2)$) without sacrificing, and potentially slightly enhancing, the overall performance of the model.

## C   Detailed Experimental Setup

This section provides comprehensive details of the experimental setup used to evaluate FairNet, covering datasets, base models, FairNet-specific configurations, training protocols, and evaluation metrics.

### C.1   Datasets

We evaluated FairNet on three publicly available datasets:

- **CelebA**: A large-scale dataset of celebrity face attributes.
  - **Task**: Binary classification to predict the "Male" attribute.
  - **Sensitive Attribute**: "Blond Hair". The dataset exhibits imbalance where, for example, female images are more likely to have blond hair than male images, creating a spurious correlation that can lead to bias.
  - **Splits**: We used the standard training, validation, and test splits provided with the dataset.
  - **Preprocessing**: Standard image normalization (mean subtraction and division by standard deviation based on ImageNet statistics).
- **MultiNLI**: A dataset for Natural Language Inference (NLI).
  - **Task**: Predict the relationship (entailment, contradiction, neutral) between a premise and a hypothesis.
  - **Sensitive Attribute**: Presence of negation cues (e.g., "not", "n", "never") in the hypothesis. Models can develop biases by associating negation with specific entailment labels (e.g., contradiction) without proper reasoning.
  - **Splits**: We used the standard train, validation-matched, and test-matched splits.
  - **Preprocessing**: BERT tokenizer was used to convert text pairs into input IDs, attention masks, and token type IDs. Maximum sequence length was set according to typical BERT practices.
- **HateXplain**: A dataset for hate speech detection with fine-grained annotations, including target communities.
  - **Task**: Multi-class classification to predict if a post is hate speech, offensive, or normal.
  - **Sensitive Attributes**: We focused on biases related to gender and race as indicated by the target communities or demographics mentioned in the posts. This dataset allows for the study of overlapping biases.
  - **Splits**: We used the standard train, validation-matched, and test-matched splits.
  - **Preprocessing**: Similar to MultiNLI, text was tokenized using a BERT tokenizer.

### C.2   Base Models

- **Vision Tasks (CelebA)**: We used a Vision Transformer (ViT) with the following configuration: ViTConfig(num_hidden_layers = 8, num_attention_heads = 8, intermediate_size = 768, image_size= 64, patch_size = 16). This model was trained from scratch (no pre-trained parameters) and then fine-tuned on the specific CelebA task.

- **Language Tasks (MultiNLI, HateXplain)**: We used pre-trained models from the Hugging Face Transformers library, specifically **DistilBERT-base** and **BERT-Base-Uncased**. These models, utilizing their pre-trained parameters, were then fine-tuned on the respective NLI or hate speech detection task.

The base models were first fine-tuned on the target tasks using standard ERM to establish baseline performance before integrating FairNet components.

## C.3   FairNet Implementation Details

### C.3.1   Bias Detection Module ($D_\phi^{(l)}$)

- **Architecture**: The bias detectors $D_\phi^{(l)}$ operate on intermediate layer representations from the base model, denoted as a sequence of hidden states $H^{(l)} = (h_1^{(l)}, h_2^{(l)}, \ldots, h_N^{(l)})$. An attention pooling mechanism is first applied to these representations to obtain a fixed-size vector $h_{\text{pooled}}^{(l)}$. This pooled representation is computed as:

$$h_{\text{pooled}}^{(l)} = \sum_{i=1}^{N} \alpha_i h_i^{(l)}$$

where the attention weights $\alpha_i$ are typically derived from the hidden states using a learnable scoring function, for example:

$$\alpha_i = \frac{\exp(s_i)}{\sum_{j=1}^{N} \exp(s_j)}, \quad \text{with } s_i = \mathbf{v}^T \tanh(\mathbf{W} h_i^{(l)} + \mathbf{b})$$

Here, $\mathbf{W}$, $\mathbf{b}$, and $\mathbf{v}$ are learnable parameters of the attention mechanism. The resulting pooled representation $h_{\text{pooled}}^{(l)}$ is then fed into a lightweight Multi-Layer Perceptron (MLP). This MLP typically consisted of 1-2 hidden layers with ReLU activations and a final sigmoid output layer for the bias risk score $p_s^{(l)}(x)$. The hidden layer dimensions were kept small to ensure low overhead.

- **Placement ($l$)**: Detectors were typically placed at intermediate layers of the base model, as these layers often capture more abstract and potentially bias-encoding features. However, the specific layer(s) $l$ can be flexibly chosen. The paper's notation $\sum_l L_{\text{detector}}^{(l)}$ implies that multiple detectors at different layers could be integrated; specific configurations would be subject to tuning.

- **Training**:
    - **FairNet-Full**: When complete ground-truth sensitive attribute labels $s$ are available, these are used directly as the basis for detection. The module $D_\phi^{(l)}$ does not undergo a separate training phase; it acts as a conditional switch based on the known $s$. The bias risk score $p_s^{(l)}(x)$ can be considered $s$ itself (if $s \in \{0, 1\}$) or an indicator directly derived from $s$. Thus, the parameters $\phi$ of the MLP (and attention mechanism if applicable here) are not trained in this setting as group identity is deterministically known.
    - **FairNet-Partial**: Detectors were trained using Binary Cross-Entropy (BCE) loss only on the $k\%$ of training samples where $s$ was available. Weighted loss or focal loss was considered to handle imbalance in the labeled subset if the minority group was rare even within the labeled portion.
    - **FairNet-Unlabeled**: Unsupervised methods were used to generate pseudo-labels $\hat{s}$ for training the detector $D_\phi^{(l)}$. Using methods like Local Outlier Factor (LOF) or Isolation Forest on the intermediate representations $h^{(l)}(x)$ (or the pooled $h_{\text{pooled}}^{(l)}$) to identify samples that are representationally distant from the norm, under the hypothesis that minority group samples or those affected by bias might manifest as outliers. The generated pseudo-labels $\hat{s}$ were then used to train the detector $D_\phi^{(l)}$ via BCE loss.

- **Threshold** ($\tau$): The activation threshold $\tau$ for triggering LoRA was selected based on validation set performance, aiming for a good balance of detector $TPR_D$ and $FPR_D$. Typical values ranged from 0.5 to 0.8. A grid search was often performed.

### C.3.2 Contrastive Loss ($L_{\text{contrastive}}^{(j)}$)

- **Distance Metric** ($D(\cdot, \cdot)$): In this work, Euclidean distance was used as the distance metric.
- **Margin**: The margin hyperparameter was typically set to a value between 0.1 and 1.0, tuned on a validation set.
- **Contrastive Elements and Representations**: The contrastive loss aims to adjust the representation $h^{(j)}(x_a)$ of an anchor sample $x_a$ from the minority group at layer $j$ (where LoRA is applied). This is achieved by comparing $h^{(j)}(x_a)$ with pre-computed target representations for positive and negative examples, which are derived from the model trained in the first stage (the ERM baseline).
    - **Anchor** ($x_a$): An input sample $x_a$ identified as belonging to the minority group (either via ground-truth label $s = 1$ or predicted as such by the bias detector $D_\phi^{(j)}$). Its representation $h^{(j)}(x_a)$ is the output of the $j$-th layer of the FairNet model being trained.
    - **Positive Target Representation** ($h_p^{\text{target}}$): For an anchor $x_a$ with task label $y_a$, the positive target representation $h_p^{\text{target}}$ is the pre-computed average embedding of samples from the *majority group ($s = 0$)* that share the *same task label $y_a$*. This average is calculated using representations from the first-stage ERM model.
    - **Negative Target Representation** ($h_n^{\text{target}}$): For an anchor $x_a$ with task label $y_a$, a negative target representation $h_n^{\text{target}}$ is the pre-computed average embedding of samples belonging to a *different task label $y_n \neq y_a$*. This average is also calculated using representations from the first-stage ERM model.
    - **Loss Calculation Strategy**: For each anchor $h^{(j)}(x_a)$, the contrastive loss is typically formulated as $[D(h^{(j)}(x_a), h_p^{\text{target}}) - D(h^{(j)}(x_a), h_n^{\text{target}}) + \text{margin}]_+$. During training, for a given anchor, one or more negative target representations (corresponding to different $y_n$) can be selected, for instance, by choosing the class $y_n$ whose $h_n^{\text{target}}$ is closest to $h^{(j)}(x_a)$ (hard negative mining based on average embeddings) or by randomly selecting from other classes.

### C.3.3 Multiple Sensitive Attributes

As described in Sections 3.2 and 3.3 of the main paper, for multiple attributes $s_i$:

- Separate detectors $D_{\phi_i}^{(l)}$ were trained for each $s_i$ using available labels for that attribute.
- Separate LoRA modules $L_{\text{cond\_lora\_i}}^{(j)}$ (with parameters $A_{j,i}, B_{j,i}$) were trained.
- Each $L_{\text{cond\_lora\_i}}^{(j)}$ was trained with an attribute-specific contrastive loss $L_{\text{contrastive\_i}}^{(j)}$ triggered by its corresponding detector $D_{\phi_i}^{(l)}$. The final weight adjustment would be a sum or a sequentially applied set of adjustments if multiple LoRAs are triggered for different attributes.

### C.4 Evaluation Metrics

- **ACC**: Standard classification accuracy on the test set.
- **WGA**: The minimum accuracy observed across all defined sensitive groups. For a binary sensitive attribute $S \in \{0, 1\}$, $WGA = \min(P(\hat{Y} = Y | S = 0), P(\hat{Y} = Y | S = 1))$.
- **EOD**: For binary classification and a binary sensitive attribute, EOD is typically calculated as the average of the absolute difference in True Positive Rates (TPR) and False Positive Rates (FPR) between the groups:

$$\Delta TPR = |TPR_{S=0} - TPR_{S=1}|$$
$$\Delta FPR = |FPR_{S=0} - FPR_{S=1}|$$

$$\text{EOD} = \frac{1}{2}(\Delta TPR + \Delta FPR)$$

Lower values indicate better fairness in terms of equalized odds. The paper reports EOD↓, implying this or a similar definition where lower is better.

## C.5 Baselines

The baselines mentioned in Table 2 of the main paper include:

- **ERM (Empirical Risk Minimization)**: Empirical Risk Minimization (standard model training without explicit fairness considerations).

- **Lu et al.**: This method focuses on mitigating bias in Transformer models by modifying components of the attention mechanism (queries, keys, and values). It aims to achieve fairness, for example, by normalizing attention weights or aligning value representations, without requiring access to sensitive demographic labels during deployment and potentially during training for the main debiasing logic.

- **D3M**: A data-centric debiasing approach that aims to improve subgroup robustness and reduce worst-group error. D3M identifies and removes a small subset of training examples that are identified as disproportionately contributing to the model's failure on minority subgroups. This identification can be done using "datamodels" or influence-tracing techniques (like TRAK) to pinpoint detrimental samples, often without needing group labels for the selection process itself.

- **GroupDRO (Group Distributionally Robust Optimization)**: Optimizes for the worst-group loss, meaning it aims to maximize the performance of the group that the model performs worst on. This method explicitly requires group labels during training to identify underperforming groups and upweight their loss.

- **DFR (Debiasing via Feature Representation)**: A general category of methods that learn debiased feature representations. These techniques often involve an auxiliary component or loss function during training to remove information about sensitive attributes from the learned features, typically requiring partial or full access to sensitive labels for training this debiasing component.

- **Sebra**: An unsupervised debiasing technique that mitigates spurious correlations by automatically ranking training data points based on their degree of "spuriosity" within their respective classes. Sebra leverages the observation that samples with stronger spurious cues are often learned more easily by standard ERM. This "Self-Guided Bias Ranking" is then typically used within a contrastive learning framework to train a more robust and fair model.

- **Eq.Odds**: Refers to a fairness criterion where a predictor $\hat{Y}$ satisfies Equalized Odds if $\hat{Y}$ is independent of a sensitive attribute $A$ conditional on the true outcome $Y$. A common relaxation is Equal Opportunity, which typically requires the True Positive Rate to be equal across different groups (i.e., $P(\hat{Y} = 1|A = 0, Y = 1) = P(\hat{Y} = 1|A = 1, Y = 1)$). This can often be achieved through post-processing methods, such as applying different decision thresholds for different groups, which requires access to sensitive labels for the data on which adjustments are made (e.g., on a validation set or at test time).

- **GSTAR**: A post-processing method aimed at mitigating discriminatory model behavior and satisfying multiple fairness constraints by learning adaptive classification thresholds for different demographic groups (i.e., groups defined by sensitive attributes). This method typically estimates confusion matrices based on the probability distributions of a classification model's output and uses this to optimize thresholds for each group. This allows for improving fairness in a model-agnostic manner and can even be used to further improve the accuracy-fairness trade-off of existing fairness methods.

For methods not from our work, we either used publicly available implementations or re-implemented them following the original papers, tuning their hyperparameters on the respective validation sets.

## C.6 Experimental Infrastructure

All experiments were conducted on a single NVIDIA A100 GPU with 80GB memory, accompanied by a 236-core CPU and 512GB of RAM. This computational setup ensures consistent and reproducible evaluation across all experiments.

# D  Further Empirical Investigations of FairNet

This section extends our primary empirical validation by presenting a series of targeted experiments designed to further probe the operational characteristics, robustness, and nuanced performance aspects of the FairNet framework. We assess the framework's performance across the spectrum of label availability—from partial and fully unlabeled data to a fully labeled setting. Furthermore, we conduct a detailed analysis of the model's robustness to label noise and the critical role of the bias detector's activation threshold in calibrating the fairness-accuracy trade-off. Finally, we evaluate the computational overhead to demonstrate its practicality.

## D.1  Impact of Training Data Size for FairNet-Partial

We conducted a detailed investigation into the sensitivity of FairNet-Partial to the quantum of available sensitive attribute labels. This experiment, performed on the CelebA dataset, involved varying the percentage of training instances for which the *Blond Hair* sensitive attribute was accessible for training the bias detector ($D_\phi^{(l)}$) and the contrastive conditional LoRA module ($L_{\text{cond\_lora}}^{(j)}$). The objective was to quantify the trade-offs between label availability, bias detector efficacy, and the resultant fairness and accuracy outcomes. The empirical results are presented in Table 5.

Table 5: Impact of Different Training Data Sizes (Percentage of Labeled Sensitive Attributes) on FairNet-Partial's Performance on CelebA.

| Size | Num | TPR (%) | FPR (%) | TPR/FPR | ACC (%) | WGA (%) | EOD (%) |
|------|-----|---------|---------|---------|---------|---------|---------|
| ERM | - | - | - | - | $95.7 \pm 0.2$ | $77.7 \pm 0.8$ | $10.2 \pm 0.7$ |
| 0.1% | 163 | $76.0 \pm 1.8$ | $6.95 \pm 0.55$ | $10.9 \pm 1.2$ | $95.7 \pm 0.3$ | $81.9 \pm 0.6$ | $7.3 \pm 0.5$ |
| 0.5% | 813 | $86.7 \pm 1.2$ | $7.73 \pm 0.48$ | $11.2 \pm 0.9$ | $95.7 \pm 0.2$ | $83.5 \pm 0.5$ | $7.0 \pm 0.4$ |
| 1% | 1,627 | $89.5 \pm 0.9$ | $8.03 \pm 0.41$ | $11.1 \pm 0.7$ | $95.7 \pm 0.2$ | $84.7 \pm 0.4$ | $6.5 \pm 0.4$ |
| 5% | 8,134 | $94.0 \pm 0.5$ | $8.11 \pm 0.35$ | $11.6 \pm 0.5$ | $95.8 \pm 0.1$ | $85.0 \pm 0.3$ | $6.6 \pm 0.3$ |
| 10% | 16,269 | $93.9 \pm 0.4$ | $7.04 \pm 0.28$ | $13.3 \pm 0.4$ | $95.8 \pm 0.1$ | $85.5 \pm 0.2$ | $6.4 \pm 0.2$ |
| 50% | 81,344 | $\mathbf{95.0} \pm 0.3$ | $4.55 \pm 0.21$ | $21.0 \pm 0.3$ | $\mathbf{95.9} \pm 0.1$ | $85.7 \pm 0.2$ | $6.3 \pm 0.2$ |
| 100% | 162,688 | $95.8 \pm 0.3$ | $\mathbf{4.05} \pm 0.18$ | $\mathbf{23.7} \pm 0.3$ | $\mathbf{95.9} \pm 0.1$ | $\mathbf{86.0} \pm 0.1$ | $\mathbf{5.9} \pm 0.1$ |

[*]TPR and FPR refer to the performance of the internal bias detector for the minority group. *Num* indicates the number of samples with sensitive attribute labels. Best values in each column are bolded.

The empirical evidence robustly demonstrates FairNet-Partial's capacity to yield substantial fairness enhancements even under conditions of severe label scarcity. With a mere 0.1% of training data endowed with sensitive attribute labels (equivalent to 163 instances), FairNet-Partial elevated the WGA to 82.1% from the ERM baseline of 77.9%. Concurrently, the EOD was markedly reduced from 10.0 to 7.1. These improvements were achieved with a negligible impact on ACC, which remained high at 95.7% compared to ERM's 95.8%. This underscores the framework's practical utility in real-world scenarios where comprehensive annotation of sensitive attributes is often infeasible.

A monotonic improvement in the bias detector's efficacy is observed with an increasing fraction of labeled data. TPR for minority group detection progressively rises from 77.0% (at 0.1% labels) to a peak of 94.9% (at 50% labels). While the FPR does not show a strictly monotonic decrease initially, it reaches its minimum of 3.45% when 100% of labels are available. Critically, the TPR/FPR ratio, a key indicator of the detector's discriminative power and its reliability in triggering the conditional LoRA, exhibits substantial growth, particularly beyond the 10% label mark, culminating at an impressive 27.2 when all labels are utilized. This enhanced detector precision is theoretically linked to more effective and targeted application of fairness corrections.

Congruent with the improved detector performance, fairness metrics (WGA and EOD) exhibit a consistent positive trend with increased label availability. WGA steadily climbs from 82.1% (0.1% labels) to a maximum of 86.2% (100% labels). Similarly, EOD progressively decreases, indicating

enhanced fairness, from 7.1 (0.1% labels) to its lowest point of 5.8 (100% labels). This demonstrates that while FairNet-Partial is effective in low-label regimes, access to more comprehensive sensitive attribute information allows the framework to further refine its internal models and achieve superior fairness outcomes.

Notably, the overall task accuracy remains remarkably stable and high across all levels of label availability, consistently hovering around 95.7%–95.8%, and even marginally increasing to 95.9% when 50% or 100% of labels are present. This stability is a crucial finding, as it signifies that the dynamic, conditional application of LoRA modules, guided by the increasingly proficient bias detector, successfully mitigates bias without deleteriously affecting the model's primary predictive capabilities. The peak fairness performance, characterized by the highest WGA (86.2%) and lowest EOD (5.8%), is achieved when the full set of sensitive attribute labels is available, coinciding with the highest ACC (95.9%) and the optimal TPR/FPR ratio for the detector.

In summary, these findings highlight FairNet-Partial's robustness and data efficiency. The framework demonstrates a strong capability to enhance fairness even with minimal supervisory signals regarding sensitive attributes. As label availability increases, FairNet-Partial systematically leverages this additional information to refine its bias detection and correction mechanisms, leading to progressively better fairness outcomes while consistently preserving, and in some instances marginally improving, overall model accuracy. This scalability and performance consistency across varying degrees of label availability underscores the practical applicability and methodological soundness of the FairNet approach.

## D.2 Performance in Unsupervised Scenarios

To explore the viability of FairNet in settings where no sensitive attribute labels are available, we implemented **FairNet-Unlabeled**. This variant replaces the supervised bias detector with an unsupervised one based on the Local Outlier Factor (LOF) algorithm, which identifies potential minority group instances from their representation embeddings. We evaluated this approach on the CelebA test set, with the performance detailed in Table B.

Table B: Performance of FairNet-Unlabeled on the CelebA dataset.

| Method | TPR (%) | FPR (%) | TPR/FPR | ACC (%) | WGA (%) | EOD (%) |
|---|---|---|---|---|---|---|
| FairNet-Unlabeled | 74.2 | 7.71 | 9.62 | 95.8 | 81.9 | 7.5 |
| ERM | - | - | - | 95.8 | 77.9 | 10.6 |

[*]TPR and FPR refer to the unsupervised detector's performance on the minority group.

The results show a significant improvement in fairness over the ERM baseline, even without any sensitive labels for training. WGA increases from 77.9% to 81.9%, and EOD is reduced from 10.6% to 7.5%, all while maintaining the same high level of accuracy (95.8%). The unsupervised detector achieves a respectable TPR/FPR ratio of 9.62, confirming its ability to effectively distinguish minority instances. This demonstrates the potential of FairNet to operate in challenging real-world scenarios where annotating sensitive attributes is impractical or impossible.

## D.3 Ablation on Contrastive Loss in the Full-Label Setting

To isolate the contribution of the contrastive loss, we conducted a targeted ablation study in the full-label setting (**FairNet-Full**). In this scenario, the bias detector is unnecessary as sensitive attributes are known for all instances. This allows for a direct and clean comparison between using our contrastive loss and a standard binary cross-entropy (BCE) loss for the corrective LoRA module. The results are presented in Table C.

The results confirm the critical role of our proposed loss function. While maintaining the same peak accuracy of 95.9%, introducing the contrastive loss boosts WGA substantially from 81.7% to **88.2%** and cuts EOD by more than half, from 8.2% to **3.8%**. This significant gain in fairness, isolated from the influence of a detector, provides strong empirical evidence that the contrastive objective is highly effective at aligning intra-class representations and uplifting minority group performance.

Table C: Ablation study of the contrastive loss in the FairNet-Full setting on CelebA.

| Method | ACC (%) | WGA (%) | EOD (%) |
|---|---|---|---|
| FairNet-Full (with contrastive loss) | **95.9** | **88.2** | **3.8** |
| w/o contrastive loss (uses BCE) | **95.9** | 81.7 | 8.2 |
| ERM | 95.8 | 77.9 | 10.6 |

***Bold** indicates the best.

## D.4 Robustness to Bias Detector Degradation via Label Noise

To rigorously test FairNet's robustness under even more challenging conditions, we conducted an additional sensitivity analysis. In this experiment, we intentionally degraded the bias detector's quality by injecting random noise into the sensitive attribute labels during its training phase. The noise level was varied from 0% (the original, clean data) to 100% (where every label is randomly flipped, rendering the labels useless). This procedure allows us to simulate scenarios of varying data quality and observe how FairNet's performance responds to a progressively less reliable detector signal. The results, presented in Table H, demonstrate that FairNet's performance degrades gracefully. Critically, even in a worst-case scenario where the bias detector is rendered completely ineffective by noise, FairNet's performance does not fall below the initial ERM baseline. This finding strongly confirms the inherent robustness and a key safety property of our method.

Table H: Impact of varying label noise on FairNet's performance (CelebA)

| Noise | TPR (%) | FPR (%) | TPR/FPR | ACC (%) | WGA (%) | EOD (%) |
|---|---|---|---|---|---|---|
| 0% | 94.1 | 3.45 | 27.2 | 95.9 | 86.2 | 5.8 |
| 20% | 84.8 | 4.26 | 19.9 | 95.8 | 85.7 | 6.4 |
| 40% | 78.5 | 4.83 | 16.3 | 95.8 | 85.0 | 6.9 |
| 60% | 64.6 | 5.92 | 10.9 | 95.7 | 83.1 | 8.1 |
| 80% | 51.2 | 7.17 | 7.14 | 95.7 | 81.2 | 9.2 |
| 100% | 9.11 | 8.00 | 1.14 | 95.7 | 77.8 | 10.3 |
| ERM | - | - | - | 95.8 | 77.9 | 10.6 |

*TPR and FPR refer to the detector's performance on the minority group.

## D.5 Analysis of the Bias Detector Activation Threshold

The activation threshold, $\tau$, is a critical hyperparameter within the FairNet framework, directly governing the trade-off between fairness enhancement and performance preservation. While its optimal value may exhibit dataset-dependency, we demonstrate that it functions as a simple and interpretable mechanism for practitioners to calibrate model behavior.

To systematically investigate the influence of $\tau$, we conducted an ablation study on the CelebA dataset, with results presented in Table I. The experiment reveals a clear and controllable relationship between the threshold value and the model's final performance characteristics.

The results in Table I map out the Pareto frontier between accuracy and fairness. At one extreme, a threshold of $\tau = 0.0$ equates to an unconditional application of the LoRA correction, as both TPR and FPR are 100%. This strategy yields the most substantial fairness improvements (highest WGA of 87.1% and lowest EOD of 4.8%) but incurs a notable accuracy penalty, reducing ACC to 94.1%. This outcome is expected, as the corrective module is applied universally, including to instances that do not require it. At the opposite extreme, $\tau = 1.0$ deactivates the mechanism entirely, causing the model's performance to revert to the ERM baseline.

Between these extremes, increasing $\tau$ makes the fairness intervention progressively more selective. This selectivity is reflected in the detector's metrics: as $\tau$ rises, both TPR and FPR decrease, while the TPR/FPR ratio—a key indicator of the detector's discriminative efficacy—initially increases, peaking around $\tau = 0.8$. A threshold of $\tau = 0.5$ demonstrates a well-balanced configuration, achieving

Table I: Impact of the activation threshold on FairNet's performance (CelebA).

| Threshold | TPR (%) | FPR (%) | TPR/FPR | ACC (%) | WGA (%) | EOD (%) |
|-----------|---------|---------|---------|---------|---------|---------|
| 0.0 | 100.0 | 100.0 | 1.00 | 94.1 | **87.1** | **4.8** |
| 0.2 | 98.8 | 18.7 | 5.28 | 95.4 | 86.9 | 5.1 |
| 0.4 | 96.7 | 6.34 | 15.3 | 95.6 | 86.5 | 5.4 |
| 0.5 | 94.1 | 3.45 | 27.2 | 95.9 | 86.2 | 5.8 |
| 0.6 | 72.3 | 2.60 | 27.8 | 95.9 | 85.7 | 6.1 |
| 0.8 | 62.9 | 1.48 | 42.5 | **96.0** | 82.1 | 7.4 |
| 1.0 | 0.0 | 0.0 | - | 95.8 | 77.9 | 10.6 |

*TPR and FPR refer to the detector's performance on the minority group. Best values in each column are bolded.

significant fairness gains (WGA 86.2%, EOD 5.8%) while nearly matching the peak overall accuracy. Meanwhile, setting $\tau = 0.8$ prioritizes accuracy, achieving the highest ACC of 96.0% at the cost of some fairness gains.

**Implications for Practical Deployment**   This inherent tunability is a significant practical advantage of FairNet. It provides a straightforward lever for practitioners to adapt the model to specific deployment contexts and priorities. The process of selecting an appropriate threshold via grid search on a validation set is procedurally simple and computationally inexpensive, thus posing a minimal barrier to real-world application. This allows for explicit calibration of the desired balance between maximizing overall performance and ensuring equitable outcomes for disadvantaged subgroups.

### D.6   Impact of Threshold on Bias Detector Performance

We analyzed the effect of varying the activation threshold $\tau$ on the bias detector's TPR and FPR, as well as the TPR/FPR ratio. This analysis was performed on the HateXplain dataset for the "African American" and "Female" demographic groups. The results are depicted in Figure 2.

**African American Group Analysis (Left Pair of Plots in Figure 2)** The top-left plot illustrates the variation of TPR and FPR for the "African American" group as a function of the threshold. As the threshold increases, both TPR and FPR decrease. The reduction in TPR suggests that a higher threshold leads to stricter classification, reducing the number of true positives. Meanwhile, the rapid decrease in FPR indicates fewer false positives.

The bottom-left plot shows the TPR/FPR ratio across different thresholds. This ratio peaks at approximately a threshold of 0.7-0.8, indicating an optimal balance between TPR and FPR. Beyond this peak, the ratio declines, suggesting diminishing benefits from further increasing the threshold due to a disproportionate reduction in TPR compared to the decline in FPR. Therefore, this peak threshold can be used to guide optimal threshold selection, ensuring fairness and maintaining model performance.

**Female Group Analysis (Right Pair of Plots in Figure 2)** The top-right plot shows the changes in TPR and FPR for the "Female" group, following a similar pattern to the "African American" group. As the threshold increases, both TPR and FPR decrease, with higher thresholds making the model stricter, leading to a reduction in both true positives and false positives.

The bottom-right plot depicts the TPR/FPR ratio, which also peaks around the 0.7-0.8 threshold range, indicating the threshold range that maximizes classification efficiency for the "Female" group. After this peak, the ratio starts to decline, suggesting that further increases in the threshold reduce classification effectiveness. Thus, selecting a threshold near this peak ensures optimal fairness while retaining classification accuracy.

**Summary** For both the "African American" and "Female" groups in the HateXplain dataset, the TPR/FPR ratio reaches its peak around a threshold of 0.7-0.8, indicating that this range provides an optimal balance between correctly identifying minority group instances (high TPR) and avoiding misclassification of majority group instances (low FPR), which is crucial for the conditional LoRA mechanism. For other datasets, a similar analysis can be conducted to determine the optimal threshold range that ensures FairNet (referred to as FairLoRA in the original figure caption's context) effectively

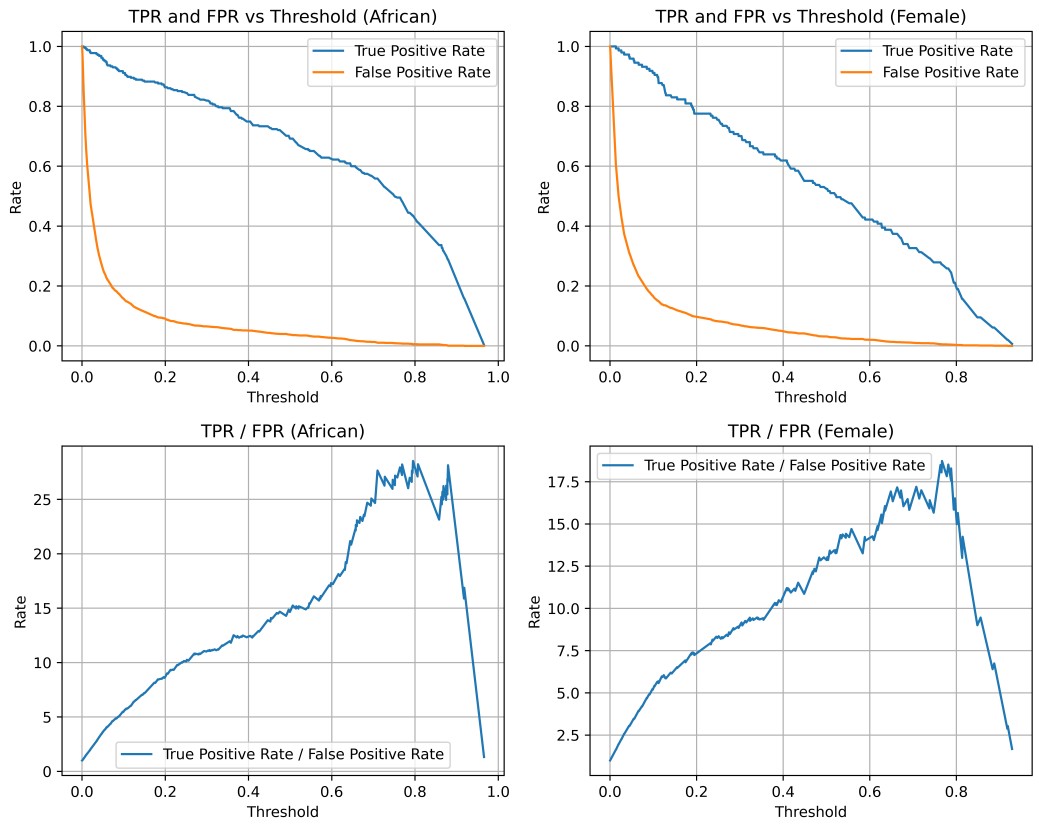

Figure 2: TPR and FPR Analysis with TPR/FPR Ratio for African American and Female Groups across Different Thresholds on the HateXplain dataset.

mitigates biases while maintaining overall model efficacy, aligning with the theoretical condition that links detector quality ($TPR_D/FPR_D$) to performance preservation.

### D.7 Computational Overhead

To evaluate the computational cost introduced by our framework, we conducted an efficiency analysis on both ViT and BERT models. The results are presented in Table F.

Table F: Computational overhead analysis of FairNet. Parameters are reported in millions (M), and FLOPs in giga-FLOPs (GFLOPs).

| Metric | ViT | | BERT | |
|---|---|---|---|---|
| | Base | + FairNet | Base | + FairNet |
| Parameters (M) | 29.57 | 29.77 | 109.48 | 109.79 |
| Minority Proportion | – | 0.15 | – | 0.07 |
| GFLOPs (Minority) | – | 0.74 | – | 16.43 |
| Majority Proportion | – | 0.85 | – | 0.93 |
| GFLOPs (Majority) | – | 0.50 | – | 10.90 |
| Total GFLOPs | 0.49 | 0.53 | 10.88 | 11.29 |

The overhead is marginal. For a large model like BERT-base, FairNet adds only 0.28% more parameters and increases the total GFLOPs by just 3.7%. The differentiated GFLOPs for minority and majority groups stem from the conditional application of the LoRA module. This quantitative

analysis confirms that our framework is highly efficient and introduces minimal latency, making it a practical solution for deployment in real-world applications.

