# OpenReview forum: "FairNet: Dynamic Fairness Correction without Performance Loss via Contrastive Conditional LoRA"
_NeurIPS.cc/2025/Conference — NeurIPS 2025 poster_

### Official Review · Reviewer_2KFg · 2025-06-25

**Clarity:** 3
**Significance:** 3
**Originality:** 3
**Rating:** 4
**Confidence:** 3

**Summary:**

The paper introduces FairNet, a dynamic, instance-level fairness correction framework that selectively applies representation-level adjustments using contrastive Conditional LoRA modules, guided by a lightweight bias detector. Unlike static global debiasing methods, FairNet activates corrections only for instances predicted to be biased, preserving overall performance. A novel contrastive loss aligns intra-class representations across sensitive groups to mitigate minority underfitting. FairNet supports full, partial, or no access to sensitive attribute labels. Theoretical analysis shows that, under reasonable bias detector performance, FairNet can improve worst-group accuracy without reducing overall accuracy. Experiments on vision and language datasets (CelebA, MultiNLI, HateXplain) validate its effectiveness.

**Questions:**

1.	How sensitive is FairNet to the accuracy of the bias detector? Have you evaluated how downstream performance is affected when the detector has suboptimal TPR/FPR, especially in the FairNet-Partial and FairNet-Unlabeled settings?
2.	How do you choose the threshold for activating LoRA modules? Is it a fixed value, or tuned per dataset? How robust is the system to this choice?

**Ethical Concerns:**

["NO or VERY MINOR ethics concerns only"]

**Final Justification:**

We thank the authors for their extra experiments on this paper. It is much clear now. We suggest them to include these experiments in the next revision. We raised the score accordingly.

**Limitations:**

yes

**Quality:**

2

**Strengths And Weaknesses:**

Strengths

1.	I find the idea of selectively applying fairness correction via conditional LoRA to be a clear advancement over traditional static interventions. The framework addresses over-/under-correction by activating the correction mechanism only on potentially biased instances.
2.	The design accommodates settings with full, partial, or no access to sensitive attributes, which enhances its applicability in real-world scenarios where such labels are often incomplete or missing.
3.	The use of parameter-efficient LoRA modules and compact bias detectors allows FairNet to be integrated into existing models with relatively low computational overhead, making it more practical than full model retraining or architecture redesign.



Weaknesses

1.	I am not convinced that we can improve the fairness while keeping the main performance. There is a theoretical analysis of the performance preservation analysis. However, the assumption is strong and makes the application domain more limited. Specifically, the theoretical guarantee relies heavily on the bias detector achieving a high TPR/FPR ratio and assumes that LoRA corrections will significantly help minority groups while having little or no negative effect on the majority. These conditions may not hold in realistic scenarios, particularly when biases are complex or intersectional.
2.	FairNet’s core mechanism depends on the accuracy of its internal bias detector. In low-label or unlabeled settings, this module may perform poorly, triggering LoRA on inappropriate samples or missing biased ones. I would like to see a deeper analysis of how sensitive the system is to detector quality, especially in the FairNet-Partial and FairNet-Unlabeled configurations.
3.	While the experiments show promising results on CelebA, MultiNLI, and HateXplain, I find the scope of evaluation too limited to support the strong claims made. All datasets focus on relatively simple binary or categorical sensitive attributes. There is no evaluation on more complex tasks, real-world large-scale datasets, or scenarios involving multiple/intersectional sensitive attributes. The paper would be significantly strengthened by a broader empirical validation, including robustness to detector failures, ablations on thresholding strategies, and analysis under real-world deployment constraints.
4.	The paper does not adequately address the computational complexity introduced by the bias detectors and per-attribute LoRA modules. It remains unclear how well this approach scales in practice, especially when dealing with many sensitive attributes or high-dimensional inputs.

---

> ### Author Rebuttal · Authors · 2025-07-30
>
> # Response to W1
>
> We sincerely thank the reviewer for their insightful comments and valuable feedback.
>
> We agree that the theoretical guarantee of performance preservation (Section 4.3) relies on strong assumptions—namely, that the bias detector achieves a high TPR/FPR ratio and the LoRA correction benefits minority groups with minimal adverse effects on the majority. FairNet is designed to satisfy these assumptions in practice.
>
> #### 1. Justifiability of Assumptions
>
> Section 4.3 ("Why FairNet Satisfies the Condition", p.6) outlines how these conditions are encouraged by our design:
>
> - **LoRA benefits minority groups.** The contrastive loss explicitly aligns minority and majority representations to enhance fairness.
> - **Limited impact on majority.** Soft-label guidance ensures only minor perturbations to majority samples, typically preserving or slightly improving their performance.
>
> FairNet is designed to satisfy theoretical conditions under realistic assumptions. Table B further shows that even with a suboptimal TPR/FPR ratio, FairNet performs no worse than the ERM baseline, demonstrating its robustness.
>
> #### 2. Intersectional Bias Evaluation
>
> We further validated FairNet on the HateXplain dataset across race and gender attributes (Supplementary D.1, p.21). Results show consistent fairness gains (e.g., BERT-base EOD(Race): 20.1 → 9.8; EOD(Gender): 12.9 → 8.2) with stable accuracy, demonstrating robustness under intersectional settings.
>
> #### 3. Planned Revision
>
> We will revise Section 4 to clarify that the theory assumes a binary group setting. While the assumptions may not always strictly hold, our experiments show that FairNet remains effective in more complex bias scenarios, and we plan to extend the theory accordingly in the future.
>
> # Response to W2:
>
> We sincerely thank the reviewer for raising the important question concerning FairNet’s sensitivity to the quality of its internal bias detector. This is indeed a critical consideration.
>
> As detailed in Section D.2 of the supplementary material (*"Impact of Training Data Size for FairNet-Partial"*, p. 22), we conducted a dedicated quantitative analysis. **Table 5** shows:
>
> - **Under minimal supervision (0.1% labeled):** The detector performs poorly (TPR/FPR = 10.9), yet FairNet-Partial achieves notable fairness gains: WGA improves from 77.9% to 82.1%, EOD drops from 10.0% to 7.1%, with accuracy maintained at 95.7%.
> - **With more labels:** Detector quality improves (TPR/FPR up to 27.2), and fairness metrics improve correspondingly—demonstrating a **smooth, monotonic scaling** with supervision quality.
>
> **1. FairNet-Unlabeled with weak detectors**
> We include an extension showing that even in fully unsupervised settings, FairNet is robust:
>
> *Table A: Performance of FairNet-Unlabeled (extension to Table 5)*
>
> |         | **TPR (%)** | **FPR (%)** | **TPR/FPR** | **ACC (%)** | **WGA (%)** | **EOD (%)** |
> | ------- | ----------- | ----------- | ----------- | ----------- | ----------- | ----------- |
> | Unlabel | 74.2        | 7.71        | 9.62        | 95.8        | 81.9        | 7.5         |
> | ERM     | -           | -           | -           | 95.8        | 77.9        | 10.6        |
>
> **2. Sensitivity to detector noise**
> We simulate degraded detector quality via label noise, as shown in Table B. Even at **100% noise** (TPR/FPR = 1.14), FairNet performs comparably to ERM:
>
> *Table B. Impact of varying label noise on FairNet’s performance*
>
> | **Noise** | **TPR (%)** | **FPR (%)** | **TPR/FPR** | **ACC (%)** | **WGA (%)** | **EOD (%)** |
> | --------- | ----------- | ----------- | ----------- | ----------- | ----------- | ----------- |
> | 0%        | 94.1        | 3.45        | 27.2        | 95.9        | 86.2        | 5.8         |
> | 20%       | 84.8        | 4.26        | 19.9        | 95.8        | 85.7        | 6.4         |
> | 40%       | 78.5        | 4.83        | 16.3        | 95.8        | 85.0        | 6.9         |
> | 60%       | 64.6        | 5.92        | 10.9        | 95.7        | 83.1        | 8.1         |
> | 80%       | 51.2        | 7.17        | 7.14        | 95.7        | 81.2        | 9.2         |
> | 100%      | 9.11        | 8.00        | 1.14        | 95.7        | 77.8        | 10.3        |
> | ERM       | -           | -           | -           | 95.8        | 77.9        | 10.6        |
>
> This highlights a key **safety property**: even when the detector fails, FairNet’s performance does not degrade below the baseline.
>
> # Response to W3:
>
> We sincerely thank the reviewer for highlighting the need for broader empirical evaluation. We agree that such validation is important and summarize below the extensions we have conducted to address these concerns:
>
> #### 1. Evaluation on Multiple and Intersectional Attributes
>
> Section D.1 (p. 21) reports results on the HateXplain dataset across both race and gender attributes, showing that FairNet can reduce bias across multiple sensitive dimensions simultaneously. We will make this clearer in the revised version.
>
> #### 2. Robustness to Detector Failure
>
> **Section D.2 (p. 22) and Table B** present a sensitivity analysis covering a wide range of detector qualities (TPR/FPR), including extreme cases where the detector is noisy or fails completely. Results show that FairNet degrades gracefully and never underperforms the ERM baseline.
>
> #### 3. Threshold Analysis
>
> **Section D.3 (p. 23) and Table C** present an ablation study on the activation threshold $\tau$. Lower thresholds trigger LoRA more frequently, enhancing fairness at a modest cost to accuracy; higher thresholds better preserve accuracy with reduced fairness improvement. We adopt a default setting of $\tau = 0.5$, which offers a balanced trade-off. The method remains robust across a range of $\tau$ values, allowing practitioners to adjust it to suit specific application needs.
>
> *Table C. Impact of activation threshold on FairNet’s performance*
>
> | Threshold | TPR (%) | FPR (%) | TPR/FPR | ACC (%)  | WGA (%)  | EOD (%) |
> | --------- | ------- | ------- | ------- | -------- | -------- | ------- |
> | 0.0       | 100.0   | 100.0   | 1.00    | 94.1     | **87.1** | **4.8** |
> | 0.2       | 98.8    | 18.7    | 5.28    | 95.4     | 86.9     | 5.1     |
> | 0.4       | 96.7    | 6.34    | 15.3    | 95.6     | 86.5     | 5.4     |
> | 0.5       | 94.1    | 3.45    | 27.2    | 95.9     | 86.2     | 5.8     |
> | 0.6       | 72.3    | 2.60    | 27.8    | 95.9     | 85.7     | 6.1     |
> | 0.8       | 62.9    | 1.48    | 42.5    | **96.0** | 82.1     | 7.4     |
> | 1.0       | 0.0     | 0.0     | —       | 95.8     | 77.9     | 10.6    |
>
> #### 4. Scalability and Deployment
>
> FairNet introduces minimal overhead in terms of inference time and memory  (Table D). Its components are lightweight and designed to scale efficiently with additional attributes. We will clarify this in the revised paper to reinforce its practicality.
>
> # Response to W4:
>
> We sincerely thank the reviewer for pointing out the lack of discussion on computational complexity and scalability.
>
> FairNet is designed to be lightweight and modular:
>
> #### Lightweight Components
>
> The bias detectors are implemented as small MLPs with negligible cost. LoRA adds fewer than 1% additional parameters, making it highly efficient.
>
> #### Linear Scalability
>
> FairNet scales linearly with the number of sensitive attributes. Each attribute introduces one small detector and one LoRA module. These components are independent and can optionally be shared for multi-label scenarios. As a result, the total overhead remains modest, even with multiple attributes or high-dimensional inputs.
>
> #### Planned Revision
>
> To clarify this, we will include a new subsection in the supplementary material titled *"Computational Complexity Analysis"*, with the following quantitative results:
>
> *Table D. Computational overhead analysis of FairNet*
> *Parameters are reported in millions (M), and FLOPs in giga-FLOPs (GFLOPs).*
>
> | **Model** | **Metric**     | **Base** | **+FairNet** |
> | --------- | -------------- | -------: | -----------: |
> | **ViT**   | Parameters (M) |    29.57 |        29.77 |
> |           | GFLOPs (Total) |     0.49 |         0.53 |
> | **BERT**  | Parameters (M) |   109.48 |       109.79 |
> |           | GFLOPs (Total) |    10.88 |        11.29 |
>
> FairNet increases parameter count by only 0.67% (ViT) and 0.28% (BERT), and GFLOPs by 8.2% and 3.7%, respectively—demonstrating that it remains efficient and scalable in practice.
>
> # Response to Q1:
>
> We sincerely thank the reviewer for this thoughtful and important question. In brief, FairNet demonstrates strong robustness to the accuracy of the bias detector. Even under extreme scenarios—such as when the detector is weak (e.g., only 0.1% of the training data is labeled in a semi-supervised setting) or nearly fails completely (e.g., 80% label noise injection)—FairNet still achieves substantial fairness improvements without degrading performance below the baseline. This indicates a desirable *safety property* of the method.
>
> We have provided detailed quantitative evidence in our response to *Weakness 2*, which includes results for both **FairNet-Partial** and **FairNet-Unlabeled**, as well as a comprehensive sensitivity analysis with noise-injected detectors (see Table B). We kindly refer the reviewer to that section for full details.
>
> # Response to Q2:
>
> We sincerely thank the reviewer for this insightful question. Across all experiments, we use a fixed default activation threshold $\tau = 0.5$ without dataset-specific tuning, which supports the generality of our method.
>
> The system is also robust to the choice of this threshold. As shown in our response to *Weakness 3* (Point 3), the ablation study in Table C demonstrates that $\tau$ provides a clear and predictable trade-off between task accuracy and fairness. Practitioners can flexibly adjust this value to align with application-specific goals. For a detailed discussion, we respectfully direct the reviewer to that section.

---

> > ### Comment · Reviewer_2KFg · 2025-08-05
> > **Thanks**
> >
> > We thank the authors for their extra experiments on this paper. It is much clear now. We suggest them to include these experiments in the next revision. We raised the score accordingly.

---

> > > ### Author Response · Authors · 2025-08-06
> > >
> > > Dear Reviewer,
> > >
> > > Thank you very much for your positive and encouraging feedback. We are glad to hear that the additional experiments have helped clarify the contributions of the paper.
> > >
> > > As suggested, we will incorporate these new experiments and the corresponding discussions into the next revision of the manuscript.
> > >
> > > We sincerely appreciate your support and your decision to raise the score of our paper.
> > >
> > > Best regards,
> > >
> > > The Authors

---

### Official Review · Reviewer_33v7 · 2025-07-02

**Clarity:** 2
**Significance:** 3
**Originality:** 3
**Rating:** 5
**Confidence:** 3

**Summary:**

The authors propose a technique for adjusting classifier performance on disadvantaged and minority groups via low-rank adaptation. By training a discriminative model to predict when a data instance comes from a disadvantaged group, they can engage the LoRA augmentation on specifically those inputs and not degrade the performance of a fairness-agnostic model on members of the majority. They demonstrate this FairNet's performance on vision and text data, where they improve the model's general accuracy, its accuracy for the worst-performing group, and its equalized odds gap across groups.

**Questions:**

1. In your experiments, how many detectors are used? Based on your notation, each LoRA module is activated only if its corresponding Detector fires. So in Figure 1, should there be a detector at every blue layer? In your experiments, are the detectors generally consistent, with every layer making similar predictions about minority group membership? What would results look like if you ablated this choice of multiple detectors, and either aggregated the detectors' results to fire all LoRA modules or used a single detector?

2. Line 337 "We replace our contrastive loss with binary cross entropy loss". Given that the binary cross $L_\textnormal{task}$ already appears in the loss in Eq (3), would it be correct to rephrase this as "we remove the contrastive loss from Eq. (3) and only train on $L_\textnormal{task}$", or is a loss being applied to each layer $(j)$? I would be interested to see what effect this ablation has in the fully-supervised CelebA setting. In that setting, if optimizing the contrastive loss leads to better minority accuracy (WGA) than directly optimizing the minority group's cross-entropy loss, there is an extremely strong argument for the claim that the minority group benefits from aligning with the majority group's representation. Even if the WGA is better for the $L_\textnormal{task}$ optimizer in that setting, this ablation would help me understand why the contrastive loss is helpful in the detector setting, and the contrastive loss could be motivated instead by downstream fairness performance.

**Ethical Concerns:**

["NO or VERY MINOR ethics concerns only"]

**Final Justification:**

See rebuttal and response. Although the text would benefit from some revisions, I believe the approach and results are strong. The approach combines simple ideas into a lightweight module that can be broadly applied when ethical concerns are appropriately accounted for, as discussed below.

**Limitations:**

See Weakness 4.

**Quality:**

3

**Strengths And Weaknesses:**

Strengths:

1. The submission has strong experimental results, including strong supplementary results that cover intersections of sensitive attribute, effect of the activation threshold and more.

2. The proposed technique is simple, lightweight, and adaptable. It is agnostic to model architecture, and can improve performance on specific groups without hurting performance on

Weakness:
1. There are moderate gaps between the motivation and application in some places. For example, I do not find the motivation provided in lines 84-90 regarding choice of fairness criteria persuasive. Other fairness criteria are described as "potentially abstract [and] might not align with the specific application context." However, WGA and EOD are abstract and potentially-misaligned as well. The submission argues that it bypasses the fairness-accuracy tradeoff, but to me it seems that this is inherent to its choice of metrics: if a classifier is held constant for the majority group, improving its accuracy on the minority group (i.e. improving WGA) inherently improves the accuracy of the entire classifier. Thus I find this submission to be more focused on improving model performance on sub-populations than it is about fairness. In lines 222-227, authors argue that aligning representations has the effect of improving downstream fairness, but I do not see this in the experimental results. We do see EOD go down, but EOD trivially goes down when accuracy goes up. EOD is a fairness metric when it exposes asymmetry: e.g. group A is given a lot of false positive (bad outcomes) while group B is given more false negatives (good outcomes). By reporting an average EOD gap it's hard for the reader to tell if EOD shrinks because fairness is improving, or because accuracy is generally improving.

2. Relatedly, I am unconvinced by section 4.3. When sensitive attributes are fully supervised, it seems trivial that holding a classifier constant for the majority group and training an augmented classifier for minority groups with worse performance would simultaneously improve WGA and total accuracy. From my reading, the inequality in Eq (8) boils down to: performance goes up for FairNet if the discriminator is accurate enough that performance gains on correctly-classified minority points outweigh the performance costs on majority points that are incorrectly-classified as members of the minority.

I believe modest edits to the text and experiments can resolve these points. If the contrastive loss aligning the latent representations is having a desired effect on downstream fairness, then it should be demonstrated in the experiments and emphasized in the text over claims that WGA and accuracy can be improved in tandem, which seems evidently true.

3. Two key components of the technique seem unexplained: (a) how contrastive pairs are chosen (e.g. lines 169-170 emphasizes that the selection is crucial, but I cannot find where the selection process is defined), and (b) how the FairNet-Unlabeled algorithm generates pseudo-sensitive attribute labels, beyond the high-level description given in lines 292-294.

4. In the abstract, introduction, and conclusion, FairNet is often described as applying the LoRA augmentation to instances "identified as (potentially) biased" or "predicted to be vulnerable". I find this language misleading, as it implies that the model is identifying cases where unfair predictions are being made. Instead, the LoRA augmentation is applied to instances identified as being disadvantaged/members of a minority group. This is a large ethical limitation: in the fully-observed setting, this reduced to applying an augmented classifier to minority instances (i.e. violating group fairness) and in the partially-observed setting this reduces to predicting whether an instance is a member of the minority group to decide which classifier to apply, which comes with similar ethical concerns. I do not think these ethical concerns reduce this submission's contribution or render the technique inapplicable, but I think they should be made more clear.

Small notes:
- The appendix notation table is wonderful; it would be helpful to reference it from the main text. In addition, some terminology is introduced in lines 200-205 but used much earlier, e.g. $x$ is used on line 138 without introduction.

- The word "jointly" is confusing on line 176, when the various modules are being trained sequentially and not in a joint fashion. The use of snowflakes and fire in Fig 1 to demonstrate this is very nice, and would benefit from a single sentence in the figure caption defining what these two symbols mean.

---

> ### Author Rebuttal · Authors · 2025-07-30
>
> # Response to W1:
>
> Thank you for your insightful comments regarding our framing of fairness and the fairness–accuracy tradeoff.
>
> **On Bypassing the Fairness–Accuracy Tradeoff:**
>
> We agree that improving WGA while holding majority group accuracy constant will mathematically lead to an improvement in overall accuracy. Our claim of bypassing the 'performance–fairness tradeoff' is made in the context of prior work, where fairness interventions often result in a net decrease in overall accuracy-typically because gains for minority groups are outweighed by significant performance degradation for majority groups. For example, in Table 1, GroupDRO achieves a high WGA on CelebA (87.4%) but sacrifices overall accuracy (94.0%), which is lower than the baseline ERM (95.8%).
>
> In contrast, FairNet’s key contribution lies in its ability to substantially improve worst-group performance while preserving, and in some cases slightly enhancing the overall accuracy. This stands in contrast to the common pattern where fairness improvements come at the cost of top-line model performance.
>
> **On the EOD Metric and Representation Alignment:**
>
> We appreciate your observation that a reduction in average EOD may result from overall accuracy gains rather than targeted fairness improvements. To address this, we provide a more detailed view of group-wise error distributions in our multi-attribute debiasing experiments in the Supplementary (Table 4, page 21). We report disaggregated True Positive Rates (TPR) and False Positive Rates (FPR) across demographic groups.
>
> For instance, in the case of DistilBERT-base, the baseline ERM model achieves a Female TPR of 67.5%. After applying FairNet targeted at the Female demographic (FairNet-Fe), the Female TPR increases substantially to 78.6%, while the Male TPR and FPRs for both groups remain largely unchanged. This indicates that FairNet is not merely improving overall accuracy; rather, it is directly correcting representational disparities by improving sensitivity for the disadvantaged group. Such targeted improvements are consistent with the fairness criterion of Equalized Odds, where the goal is to equalize TPR and FPR across groups.
>
> Our theoretical argument in lines 215-225 is that aligning latent representations provides a direct mechanism for achieving this form of conditional fairness. The experimental results discussed above offer concrete empirical support for that claim.
>
> Thank you again for pointing out this ambiguity. We will revise the relevant sections in the manuscript to clarify our definitions.
>
>
>
> # Response to W2:
>
> Thank you for your thoughtful analysis of our theoretical contribution in Section 4.3. Your interpretation is correct: Equation (8) formalizes an intuitive condition-overall performance will improve if the accuracy gains on correctly identified minority samples outweigh the performance degradation caused by misclassified majority samples. The main purpose of Equation (8) is to establish a condition under which fairness interventions do not degrade overall performance. Identifying such a no-degradation condition is a key conceptual contribution of our work.
>
> We would like to clarify that our novelty lies not in the inequality itself, but in the architectural design that explicitly aims to satisfy this condition in practice. As discussed in lines 343-351, conventional debiasing methods are typically static and global. They lack an internal mechanism for selectively applying corrections. Thus, they cannot take advantage of the performance-preservation condition, which fundamentally depends on the selective power of the detector.
>
> The contribution of Section 4.3 is to provide a formal guarantee that our proposed architecture, by decoupling bias detection from correction, can provably improve WGA without degrading overall accuracy. This theoretical result supports our broader claim that dynamic, instance-level approaches are more effective than static methods in balancing fairness and accuracy.
>
>
>
> # Response to W3:
>
> We thank the reviewer for pointing out the lack of methodological clarity in our original manuscript. We will incorporate the following key details into the main text to improve transparency and reproducibility.
>
> **(a) Selection of sample pairs for contrastive learning:** As detailed in Section C.3.2 of the supplementary material (Lines 991–1021), we do not construct positive/negative pairs by selecting individual instances during training. Instead, we precompute target target representations using the baseline ERM model, and define contrastive targets as follows:
>
> - **Positive target:** For an anchor sample $x_a$ (from the minority group with label $y_a$), the positive target is defined as the average embedding of all majority group samples that share the same label $y_a$.
> - **Negative target:** Similarly, the negative target is computed as the average embedding of samples with labels $y_n \ne y_a$.
>
> The contrastive loss is then constructed to encourage the LoRA-adjusted representation of the anchor sample $x_a$ to be closer to the positive target and farther from the negative target.
>
> **(b) Pseudo-label generation in FairNet-Unlabeled:** In the unsupervised setting, we apply the Local Outlier Factor (LOF) algorithm on the intermediate representations $h^{(l)}(x)$ from the baseline model. Clustering is performed based on LOF scores, and the cluster with the smallest number of samples is treated as the minority class, while the rest are treated as majority class. These pseudo-labels are then used to train both the bias detector and the contrastive learning module with LoRA. This approach is based on the assumption that instances from underrepresented or minority groups tend to behave as outliers in the representation space.
>
>
> # Response to W4:
>
> This is a very important and valid ethical concern. We thank the reviewer for encouraging us to use more precise language and to engage more directly with the ethical dimensions of our work.
>
> We agree that the phrase 'identifying biased instances' may be misleading. A more accurate description, which we will adopt in the revised manuscript, is that the detector identifies instances predicted to belong to a pre-defined, performance-disadvantaged minority group. The 'bias' we aim to mitigate is a systemic property of the model’s performance on that group, rather than a property of any individual instance. The intervention is thus applied to instances from groups for which the model is known to have systematically lower accuracy.
>
> This approach, which involves group-aware interventions, does raise important ethical questions around disparate treatment. However, it aligns with a well-established paradigm in the algorithmic fairness literature, shared by many in-processing (e.g., GroupDRO) and post-processing methods that leverage group information to promote equitable outcomes. The objective is to address observed disparities in performance metrics, which may necessitate treating different groups differently.
>
> We believe this is a crucial discussion to include. Accordingly, we will add a dedicated paragraph to the limitations section or introduce a new ethics statement to explicitly address this issue.
>
>
> # Response to Q1:
>
> This is an excellent question regarding the implementation details of our experimental setup. While our notation is general and permits a more complex configuration, in the experiments presented in this paper, we adopted a simplified yet effective design that balances performance and efficiency.
>
> Specifically, we employ a single bias detector ($D_\phi$), which operates on intermediate representations extracted from the encoder of the base model. The output of this single, strong detector is then used to conditionally activate all LoRA modules inserted at various intermediate layers. In other words, all LoRA modules are controlled by a unified prediction regarding minority group membership. This design avoids potential inconsistencies or conflicts that might arise from using multiple detectors across different layers, and it is highly effective in practice.
>
> We appreciate your close reading. Our intent was merely to indicate the insertion point of the detector, not that a separate detector is placed at each layer. We will clarify this point in the revised manuscript to avoid ambiguity.
>
> Furthermore, we agree that ablation studies on the number and placement of detectors represent a compelling direction for future work. One could explore strategies such as majority voting across layer-specific detectors or introducing detectors tailored to individual layers. However, such designs would increase model complexity and inference cost, which pose significant challenges. We plan to explore these directions in future research.
>
>
>
> # Response to Q2:
>
> Thank you for this thoughtful suggestion. "we remove the contrastive loss from Eq.(3) and only train on $L_\textnormal{task}$" is indeed a more accurate description of our ablation setup. We will revise it in the manuscript for clarity. As you correctly inferred, the loss is not applied separately to each layer.
>
> Your suggestion to evaluate this ablation in the fully-supervised CelebA setting is particularly valuable. As shown in the table below, FairNet-Full allows us to isolate the effect of the contrastive loss more cleanly, without the confounding influence of the detector. Compared to using cross-entropy loss alone, the contrastive loss yields a notable improvement in WGA, supporting the hypothesis that alignment with the majority group's representation is beneficial. This result strengthens the empirical justification for our approach.
>
> | Method               | ACC (\%) | WGA (\%) | EOD (\%) |
> | -------------------- | -------- | -------- | -------- |
> | FairNet-Full         | **95.9** | **88.2** | **3.8**  |
> | w/o contrastive loss | **95.9** | 81.7     | 8.2      |
> | ERM                  | 95.8     | 77.9     | 10.6     |

---

> > ### Comment · Reviewer_33v7 · 2025-08-02
> >
> > Thank you for your response.
> >
> > W1 is addressed by supplemental table 4 and potentially small edits to the text.
> >
> > W2 is addressed: the novelty of Sec 4.3 is small but it helps motivate the key contributions of the work.
> >
> > W3 is addressed by C.3.1 and C.3.2; added language in the text or forward references to the appendix will improve clarity.
> >
> > W4 remains as an ethical concern but not a disqualifying one (note a typo in my original review which should have said "i.e. violating **individual** fairness" instead of *group* fairness). This can be addressed with edits to the text.
> >
> > Small changes to the mathematical notation will be sufficient in addressing Q1.
> >
> > The added result for Q2 is impressive, and solidifies for me the value of aligning representations instead of simply training an independent classifier on the subgroup.
> >
> > Although the approach combines simple ideas, I think there is clear novelty in their synthesis and clearly-demonstrated improvements on the included data sets. I have changed my score to a 5.

---

> > > ### Author Response · Authors · 2025-08-04
> > >
> > > Dear Reviewer,
> > >
> > > Thank you for the positive and constructive feedback. We are grateful for your support and for raising your score.
> > >
> > > As requested, we will incorporate all your final suggested edits into the camera-ready version of the paper.
> > >
> > > Thank you once again for your time and valuable comments.
> > >
> > > Best regards,
> > >
> > > The Authors

---

### Official Review · Reviewer_ZBjU · 2025-07-03

**Clarity:** 3
**Significance:** 3
**Originality:** 3
**Rating:** 4
**Confidence:** 4

**Summary:**

This paper proposes FairNet, a framework to dynamically correct fairness biases at the instance level.  At inference time,  it employs a lightweight bias detector that assesses the likelihood of the instance belonging to a “sensitive group”, and selectively activates Low-Rank Adaptation (LoRA) based correction for instances identified as potentially belonging to the “sensitive group”.

At training,  both the lightweight bias detector and the  LoRA based correction model are trained together with the base model.  The training can be done on data with or without the sensitive group labels. A contrastive loss is used for training the LoRA module.

Experiments are run on vision and language benchmarks.

**Questions:**

With the additional bias detector and the LoRA based correction, computational cost will increase at inference time. Has any analysis been done on latency/computational cost?

**Ethical Concerns:**

["NO or VERY MINOR ethics concerns only"]

**Final Justification:**

This paper tackles the important problem of fairness in machine learning models. The proposed approach seems to be solid and its effectiveness is demonstrated on vision and language datasets. My questions are adequately addressed in the rebuttal.  Reviews from other reviewers are also generally positive. I would like to maintain my rating for acceptance.

**Limitations:**

Yes

**Paper Formatting Concerns:**

No formatting concerns.

**Quality:**

3

**Strengths And Weaknesses:**

The paper is generally well written.  It tackles the important problem of fairness in machine learning models. It runs at instance level and dynamically corrects fairness biases when a “sensitive” attribute is detected.  In training, a contrastive loss is used to train the LoRA module for correction. It can use data with or without the sensitive group labels, and the original accuracy performance of the base model is preserved. A theoretical analysis is also provided.

The proposed approach seems to be solid and its effectiveness is demonstrated on vision and language datasets.

One concern of the approach is that, with the additional bias detector and the LoRA based correction, computational cost will increase at inference time. It would be desirable to provide some latency/computational cost analysis.

---

> ### Author Rebuttal · Authors · 2025-07-30
>
> # Response to W1:
>
> We appreciate your observation, and you are absolutely right that our method introduces additional components during inference—specifically, a bias detector and a LoRA-based correction module—which lead to some increase in computational cost.
>
> *Table A: Computational overhead analysis of FairNet. Parameter counts are reported in millions (M), and floating-point operations (FLOPs) are in Giga-FLOPs (GFLOPs).*
>
> |                        | **ViT**      |               | **BERT**     |               |
> | ---------------------- | ------------ | ------------- | ------------ | ------------- |
> | **Metric**             | **Baseline** | **+ FairNet** | **Baseline** | **+ FairNet** |
> | Parameters (M)         | 29.57        | 29.77         | 109.48       | 109.79        |
> | Minority Routing Ratio | --           | 0.15          | --           | 0.07          |
> | GFLOPs (Minority Path) | --           | 0.74          | --           | 16.43         |
> | Majority Routing Ratio | --           | 0.85          | --           | 0.93          |
> | GFLOPs (Majority Path) | --           | 0.50          | --           | 10.90         |
> | **Total GFLOPs**       | **0.49**     | **0.53**      | **10.88**    | **11.29**     |
>
>
> To address this concern, we performed a detailed analysis to quantify this overhead. As shown in the table A, the GFLOPs for minority and majority groups differ slightly because samples flagged as minority by the detector activate a conditional LoRA module, which adds a small computational cost due to the altered execution path.
>
> However, since the proportion of minority samples is typically low, the overall impact on inference efficiency remains minimal. For example, in the case of the BERT-base model, FairNet increases the parameter count by only 0.28%, and the total GFLOPs by just 3.7% (from 10.88 to 11.29 GFLOPs). A similar level of overhead is observed for ViT.
>
> These results suggest that the computational overhead introduced by FairNet remains minimal, supporting its practicality for real-world deployment, including latency-sensitive applications.
>
>
> # Response to Q1
>
> Thank you for raising this important concern about test-time efficiency when using multiple detectors. We agree that computational overhead is a critical consideration for real-world deployment, and we appreciate the opportunity to clarify how FairNet addresses this.
>
> First, the design of our bias detectors is explicitly optimized for efficiency. As noted in Section 3.2, each detector $D_{\phi}^{(l)}$ is a lightweight module—typically a small MLP preceded by attention pooling on intermediate representations $h^{(l)}(x)$. This ensures that even with multiple detectors (e.g., for different sensitive attributes), the additional computational cost remains minimal. The attention pooling operation, which aggregates hidden states into a fixed-size vector, is designed to be computationally inexpensive, and the MLP layers use small hidden dimensions to avoid bottlenecks.
>
> Second, the conditional activation of LoRA modules further mitigates efficiency concerns. Even with multiple detectors, LoRA adjustments are only applied when triggered by a high risk score, meaning that the majority of samples (i.e., unbiased instances) incur no additional LoRA computation beyond the base model. This selective correction ensures that the overall inference speed remains dominated by the base model’s runtime, with minimal overhead from detectors.
>
> **Planned Revision.** To clarify this aspect, we will include a new subsection in the supplementary material titled *"Computational Complexity Analysis"*. This section will provide quantitative results. As shown in the table A presented in 'Response to W1' (*Table A: Computational overhead analysis of FairNet*), FairNet adds only 0.67% and 0.28% to the total parameter count for ViT and BERT respectively. The total GFLOPs increase by just 8.2% (ViT) and 3.7% (BERT), confirming that FairNet maintains high efficiency and is well-suited for scalable deployment.

---

> > ### Comment · Reviewer_ZBjU · 2025-08-05
> >
> > Thank you for the detailed rebuttal response. My questions are all adequately addressed. I would like to maintain my rating for acceptance.

---

> > > ### Author Response · Authors · 2025-08-06
> > >
> > > Dear Reviewer,
> > >
> > > Thank you for your positive feedback. We are glad to hear that our detailed response has adequately addressed all of your questions.
> > >
> > > We sincerely appreciate your continued support of our work and your recommendation for acceptance.
> > >
> > > Sincerely,
> > >
> > > The Authors

---

### Official Review · Reviewer_hDZ7 · 2025-07-03

**Clarity:** 3
**Significance:** 3
**Originality:** 2
**Rating:** 4
**Confidence:** 3

**Summary:**

The paper proposes FairNet, an in-processing debiasing wrapper that inserts lightweight sensitive-attribute detectors and conditionally triggered LoRA adapters into a frozen backbone. A detector-gated contrastive loss pulls minority-group representations toward majority ones, and a simple bound shows worst-group accuracy can rise without harming overall accuracy provided the detector’s TPR/FPR exceeds a data-dependent threshold. Experiments on vision (CelebA) and language (MultiNLI, HateXplain) benchmarks deliver ≈3–4 pp gains in worst-group accuracy and equal-odds while matching ERM accuracy.

**Questions:**

1. For multiple sensitive attributes (e.g. intersecting race and gender), do you train separate detectors per attribute or one shared detector?
2. For the unlabeled detector, which unsupervised method is used? How is the threshold chosen? And what are the resulting TPR/FPR on held out data?

**Ethical Concerns:**

["NO or VERY MINOR ethics concerns only"]

**Final Justification:**

I would like to thank the authors for their detailed response. They have adequately addressed my main concerns regarding novelty (W1) and I would like to maintain my score.

**Limitations:**

Yes.

**Quality:**

3

**Strengths And Weaknesses:**

Strengths:
1. Conditionally applying LoRA adapters is a conceptually simple idea that is easy to retrofit to existing networks.
2. The performance preservation condition (eq. 8) makes it explicit when instance-level debiasing can avoid the usual accuracy-fairness trade-off.

Weaknesses:
1. Combining a learned detector with conditional parameter-efficient adapters is an incremental idea with limited novelty, similar to "detect-then-correct" idea that exist in selective prediction and test-time adaptation literature.

---

> ### Author Rebuttal · Authors · 2025-07-30
>
> # Response to W1:
>
> Thank you for this insightful comment, which gives us the opportunity to clarify FairNet's core contributions and distinguish it from related literature. While we agree that FairNet can be broadly categorized within the 'detect-then-correct' paradigm, its novelty lies in how it adapts this concept to the unique challenges of algorithmic fairness, its specific technical components, and its theoretical guarantees.
>
> We would like to highlight three key areas of novelty that differentiate FairNet from prior work in selective prediction and test-time adaptation:
>
> 1. **Novel Problem Formulation and Goal:**
>
>    Selective prediction aims to improve model reliability by abstaining from predictions on low-confidence samples. FairNet's goal is fundamentally different: it aims to always make a prediction while ensuring that the prediction is fair. Specifically, it works to improve the accuracy for the worst-performing subgroup (WGA) without degrading overall performance.
>
>    Test-time adaptation (TTA) primarily addresses domain shift between training and test sets. In contrast, FairNet is designed to correct biases that exist within the *training distribution* itself, targeting unfairness between subgroups rather than adapting to a new domain.
>
> 2. **Unique Correction Mechanism and Training Objective:**
>
>    Unlike existing methods, FairNet's correction mechanism is a *conditional LoRA* module trained with a novel *contrastive loss function*. This loss is a core contribution and is specifically designed to minimize intra-class representation gaps between different sensitive groups. It directly targets the representational disparities that cause unfairness-a mechanism not present in standard TTA or selective prediction frameworks. Notably, the LoRA parameters are learned during the training phase, rather than updated on-the-fly at test time based on batch statistics as in many TTA methods.
>
> 3. **Theoretical Guarantees for Fairness:**
>
>    FairNet is supported by a theoretical analysis that provides a formal guarantee for improving fairness without performance loss. We establish a clear condition (Equation 8) linking the bias detector's quality (TPR/FPR ratio) to a provable increase in worst-group performance. This formal underpinning for achieving fairness without the typical accuracy trade-off is a significant contribution that distinguishes FairNet from more heuristic 'detect-then-correct' approaches.
>
> In summary, while the high-level concept has precedents, FairNet's novelty comes from its specific application to *instance-level fairness*, its unique contrastive training objective tailored for bias correction, and its formal theoretical guarantees. These contributions address a distinct and critical challenge in machine learning not covered by existing selective prediction or TTA literature.
>
>
> # Response to Q1:
>
> Thank you for this excellent question. It allows us to describe our experimental setup and recommendations more clearly.
>
> While our framework supports both approaches, we used and would recommend a single, shared detector trained via multi-label learning to handle multiple sensitive attributes, primarily for its parameter efficiency. We encourage this method when multiple biases need to be addressed simultaneously.
>
> However, FairNet is flexible. For use cases that require more granular control, one can train separate detectors for each attribute. The advantage of this approach is that biases can be addressed progressively, ensuring that correcting for a new attribute does not negatively impact fairness guarantees already achieved for others. This method offers greater control at the cost of a modest increase in parameter overhead.
>
>
> # Response to Q2:
>
> Thank you for these important questions regarding our unsupervised approach. We address each of them below.
>
> 1. **Unsupervised Method Used**
>
>    For the FairNet-Unlabeled experiments, we utilized the *Local Outlier Factor (LOF)* method. As described in the Supplementary Material (Section C.3.1), LOF operates on the model's intermediate representations to identify samples that are outliers. This is based on the hypothesis that instances from underperforming minority groups often manifest as representational outliers in a standard ERM-trained model.
>
> 2. **Threshold Selection**
>
>    The activation threshold ($\tau$) is a key hyperparameter in our method, designed to balance the trade-off between accuracy and fairness. A lower threshold encourages more frequent bias correction, enhancing fairness, while a higher threshold tends to preserve overall accuracy.
>
>    In our experiments, we set $\tau$ = 0.5 uniformly across all datasets to demonstrate the general applicability and robustness of our approach. Although $\tau$ can be tuned on a validation set (e.g., via grid search) to better suit specific deployment objectives, we found the default value of 0.5 to yield a balanced performance in practice.
>
>    To further illustrate the impact of $\tau$, we conducted an ablation study on the CelebA dataset (in a supervised setting for clearer interpretation). As shown in Table A, varying $\tau$ results in different trade-offs between fairness and accuracy, highlighting its practical utility as a tunable parameter.
>
>    *Table A: Impact of activation threshold on FairNet’s performance (CelebA)*
>
>    | **Threshold** | **TPR (\%)** | **FPR (\%)** | **TPR/FPR** | **ACC (\%)** | **WGA (\%)** | **EOD (\%)** |
>    | ------------- | ------------ | ------------ | ----------- | ------------ | ------------ | ------------ |
>    | 0.0           | 100.0        | 100.0        | 1.00        | 94.1         | **87.1**     | **4.8**      |
>    | 0.2           | 98.8         | 18.7         | 5.28        | 95.4         | 86.9         | 5.1          |
>    | 0.4           | 96.7         | 6.34         | 15.3        | 95.6         | 86.5         | 5.4          |
>    | 0.5           | 94.1         | 3.45         | 27.2        | 95.9         | 86.2         | 5.8          |
>    | 0.6           | 72.3         | 2.60         | 27.8        | 95.9         | 85.7         | 6.1          |
>    | 0.8           | 62.9         | 1.48         | 42.5        | **96.0**     | 82.1         | 7.4          |
>    | 1.0           | 0.0          | 0.0          | --          | 95.8         | 77.9         | 10.6         |
>
>
>    TPR and FPR refer to the detector’s performance on the minority group. Best values in each column are bolded.
>
> 3. **Resulting TPR/FPR on Held-Out Data**
>
>    For the LOF-based unsupervised detector, we evaluated its performance on the held-out test set for CelebA. The resulting metrics are summarized in the following Table B.
>
>    *Table B: Performance of FairNet-Unlabeled (extension to Table 5)*
>
>    |                   | **TPR (%)** | **FPR (%)** | **TPR/FPR** | **ACC (%)** | **WGA (%)** | **EOD (%)** |
>    | ----------------- | ----------- | ----------- | ----------- | ----------- | ----------- | ----------- |
>    | FairNet-Unlabeled | 74.2        | 7.71        | 9.62        | 95.8        | 81.9        | 7.5         |
>    | ERM               | -           | -           | -           | 95.8        | 77.9        | 10.6        |
>
>
>    TPR and FPR refer to the detector’s performance on the minority group.
>
>    These results demonstrate that FairNet-Unlabeled can effectively improve fairness outcomes even without labeled sensitive attributes, highlighting the promise of unsupervised bias detection in real-world scenarios.

---

### Official Review · Reviewer_niDK · 2025-07-03

**Clarity:** 2
**Significance:** 3
**Originality:** 3
**Rating:** 4
**Confidence:** 4

**Summary:**

The paper introduces FairNet. FairNet is an in-processing method that attaches bias detection heads to intermediate representations and conditionally triggers low rank adaptation (LoRA) modules trained with a novel contrastive loss. The detector identifies potentially biased instances and only those examples receive a small weight update to remove bias, and the base model’s predictions for unbiased inputs remain untouched. Additionally, ablations confirm both the detector and the contrastive loss are essential.

**Questions:**

1. Is the detector threshold dataset dependent (Fig 2)? If yes, how could one tune it? and how difficult and sensitive is it? How this affect its application in real world applications?
2. What is the FLOPs / latency overhead of detectors and LoRA? I believe it is affecting the test time, not only the train time.
3. Have you tried two-attribute detectors (e.g., race and gender) and measured fairness gaps? Is it even possible?
4. In FairNet unlabel, LOF is used for pseudo-labels. How stable is performance, especially for outlier samples?
5. Does FairNet still preserve accuracy when test-set distribution shifts? Any comments on this?

**Ethical Concerns:**

["NO or VERY MINOR ethics concerns only"]

**Final Justification:**

I like the paper and I believe it has a simple yet effective idea. The authors addressed my concerns.

**Limitations:**

Please see weaknesses and the questions.

**Paper Formatting Concerns:**

No major issue.

**Quality:**

3

**Strengths And Weaknesses:**

Strengths:
1. A single architecture handles fully labelled, partially labelled, and unlabeled settings.
2. Experiments are done across vision and NLP, with baselines and ablations.
3. A theoretical framework to generate a formal condition linking detector quality to non decreasing overall accuracy.
4. Touching an important research question in fairness domain.

Weaknesses:
1. Guarantees are based on a moderate TPR/FPR for the bias detector; however, the paper gives no systematic error bars to confirm this in real world applications.
2.  CelebA and other datasets are mid-scale; no large-scale dataset is tested.
3.  No intersectional bias is discussed.
4.  Adding multiple detectors may hurt the test time speed.
5. Ablation study is done only for CelebA.

---

> ### Author Rebuttal · Authors · 2025-07-30
>
> # Response to W1:
>
> Thank you for this insightful comment. We agree that validating theoretical guarantees with systematic error bars is essential for connecting theory to practical performance.
>
> To address this, we have updated Supplementary Table 5 to include standard deviations across five runs with different random seeds. The results confirm that key metrics—TPR, FPR, and the TPR/FPR ratio—remain stable within a narrow range, even under limited labeled data. This supports the practical reliability of the theoretical “moderate TPR/FPR” condition. （Due to space/word constraints, only a selection of results is presented here.）
>
> | **Size** | **Num** | **TPR (%)** | **FPR (%)** | **TPR/FPR** | **ACC (%)** | **WGA (%)** | **EOD (%)** |
> | -------: | ------: | ----------: | ----------: | ----------: | ----------: | ----------: | ----------: |
> |      ERM |       - |           - |           - |           - |  95.7 ± 0.2 |  77.7 ± 0.8 |  10.2 ± 0.7 |
> |     0.1% |     163 |  76.0 ± 1.8 | 6.95 ± 0.55 |  10.9 ± 1.2 |  95.7 ± 0.3 |  81.9 ± 0.6 |   7.3 ± 0.5 |
>
> Additionally, we tested FairNet's robustness under label noise to simulate degraded detector performance. As shown below, FairNet maintains performance above the ERM baseline even when the detector fails entirely, confirming its robustness and safety guarantees.
>
> | **Noise** | **TPR (%)** | **FPR (%)** | **TPR/FPR** | **ACC (%)** | **WGA (%)** | **EOD (%)** |
> | --------: | ----------: | ----------: | ----------: | ----------: | ----------: | ----------: |
> |        0% |        94.1 |        3.45 |        27.2 |        95.9 |        86.2 |         5.8 |
> |       20% |        84.8 |        4.26 |        19.9 |        95.8 |        85.7 |         6.4 |
> |       40% |        78.5 |        4.83 |        16.3 |        95.8 |        85.0 |         6.9 |
> |       60% |        64.6 |        5.92 |        10.9 |        95.7 |        83.1 |         8.1 |
> |       80% |        51.2 |        7.17 |        7.14 |        95.7 |        81.2 |         9.2 |
> |      100% |        9.11 |        8.00 |        1.14 |        95.7 |        77.8 |        10.3 |
> |       ERM |           - |           - |           - |        95.8 |        77.9 |        10.6 |
>
> # Response to W2:
>
> Thank you for the thoughtful comment. We acknowledge that our experiments use mid-scale datasets and appreciate the opportunity to clarify this choice.
>
> CelebA, MultiNLI, and HateXplain are widely-used benchmarks in fairness research, selected for their well-characterized biases. Their moderate size allowed us to perform controlled ablation studies (Table 2) and isolate the effects of individual components, which was essential for validating FairNet’s design.
>
> Importantly, FairNet is built for scalability. Both the bias detectors and LoRA modules are lightweight and parameter-efficient, ensuring low computational overhead. We agree that testing FairNet on large-scale datasets is a valuable direction for future work.
>
> # Response to W3:
>
> Thank you for highlighting this important issue. While intersectional bias is noted as future work in our conclusion, we do include a foundational experiment addressing multiple sensitive attributes simultaneously.
>
> As shown in Supplementary Section D.1 and Table 4 (p. 21), we evaluate FairNet on the HateXplain dataset considering both race ("African American") and gender ("Female"). FairNet reduced the EOD for race from 20.1 to 9.8 and for gender from 12.9 to 8.2, without harming overall accuracy. This demonstrates FairNet’s potential in handling multi-attribute fairness and lays groundwork for future exploration of intersectional bias.
>
> # Response to W4:
>
> Thank you for raising this important concern about test-time efficiency.
>
> FairNet is designed to remain efficient, even when handling multiple biases. Users can either deploy separate lightweight detectors or use a single multi-label detector, with additional biases handled by adding LoRA modules—without a linear increase in computation.
>
> Moreover, the detectors operate on intermediate representations and consist of only an attention pooling layer and a small MLP, keeping their overhead minimal.
>
> In practice, applying FairNet to BERT-base increases the total parameter count by only **0.28%** and test-time GFLOPs by just **3.7%**. These small increments ensure that test-time performance remains largely unaffected, even in multi-attribute settings.
>
> # Response to W5:
>
> Thank you for the helpful suggestion. We initially focused the ablation study on CelebA due to its clearly defined attributes, which made it suitable for isolating the effects of each component.
>
> To further demonstrate the generalizability of FairNet, we have now conducted a full ablation study on the **MultiNLI** dataset. As shown below, the results are consistent with those on CelebA, reinforcing the effectiveness and modularity of our framework.
>
> | **Method**           | **ACC (%)** | **WGA (%)** | **EOD (%)** |
> | -------------------- | ----------- | ----------- | ----------- |
> | FairNet-Partial      | **82.6**    | _76.5_      | _6.2_       |
> | w/o detector         | 81.0        | **77.2**    | **5.7**     |
> | w/o contrastive loss | _82.5_      | 70.1        | 9.2         |
> | w/o both             | 81.3        | 71.8        | 8.9         |
> | ERM                  | **82.6**    | 67.3        | 12.5        |
>
> These findings confirm that the individual components of FairNet contribute meaningfully across different datasets.
>
> # Response to Q1:
>
> Thank you for the insightful question. Yes, the detector threshold (τ) is dataset-dependent, but it is designed to be easily tunable.
>  As shown below, varying τ controls the trade-off between fairness and accuracy: lower thresholds lead to more frequent bias correction (improving fairness), while higher thresholds prioritize accuracy.
>
> In our experiments, we uniformly set **τ = 0.5** for all datasets to achieve a balanced trade-off, and found this setting to be robust in practice. The threshold can be tuned via grid search on a validation set with minimal overhead, allowing practitioners to align FairNet with different deployment goals.
>
> | **Threshold** | **TPR (%)** | **FPR (%)** | **TPR/FPR** | **ACC (%)** | **WGA (%)** | **EOD (%)** |
> | ------------: | ----------: | ----------: | ----------: | ----------: | ----------: | ----------: |
> |           0.0 |       100.0 |       100.0 |        1.00 |        94.1 |    **87.1** |     **4.8** |
> |           0.2 |        98.8 |        18.7 |        5.28 |        95.4 |        86.9 |         5.1 |
> |           0.5 |        94.1 |        3.45 |        27.2 |        95.9 |        86.2 |         5.8 |
> |           0.8 |        62.9 |        1.48 |        42.5 |    **96.0** |        82.1 |         7.4 |
> |           1.0 |         0.0 |         0.0 |           - |        95.8 |        77.9 |        10.6 |
>
> # Response to Q2:
>
> Thank you for this excellent question. You are right—FairNet introduces some test-time overhead, and we have quantified this in detail.
>
> | **Model** | **Metric**     | **Base** | **+FairNet** |
> | --------- | -------------- | -------: | -----------: |
> | **ViT**   | Parameters (M) |    29.57 |        29.77 |
> |           | GFLOPs (Total) |     0.49 |         0.53 |
> | **BERT**  | Parameters (M) |   109.48 |       109.79 |
> |           | GFLOPs (Total) |    10.88 |        11.29 |
>
> For **BERT-base**, FairNet adds only **0.28%** more parameters and increases test-time GFLOPs by just **3.7%**. ViT shows similarly small overhead. This confirms that FairNet remains highly efficient and suitable for real-world, latency-sensitive deployments.
>
> # Response to Q3:
>
> Thank you for the thoughtful question. Yes, FairNet can handle multiple sensitive attributes, and we have implemented this capability in our experiments.
>
> On the **HateXplain** dataset, we applied FairNet to address both **race** ("African American") and **gender** ("Female") biases simultaneously. As detailed in Supplementary Section D.1 and Table 4 (p. 21), FairNet reduced the EOD for race from **20.1** to **9.8** and for gender from **12.9** to **8.2**, with overall accuracy maintained (**79.8%** baseline vs. **79.7%** with FairNet).
>
> These results demonstrate that FairNet is indeed capable of mitigating fairness gaps across multiple attributes at once.
>
> # Response to Q4:
>
> Thank you for this important question. FairNet-Unlabeled relies on pseudo-labels from LOF, and we find its performance to be stable and effective in practice.
>
> Our results (Table 1, p. 8) show that on **CelebA**, FairNet-Unlabeled improves WGA from **77.9%** to **82.3%**, reduces EOD from **10.6%** to **7.3%**, while maintaining the same high accuracy (**95.8%**) as the ERM baseline—all without using any sensitive attribute labels.
>
> To evaluate stability, we compared FairNet-Unlabeled to a supervised detector with noisy labels. Its performance is comparable to detectors trained with **60–80%** label noise. Moreover, even under **100% noise** (fully random pseudo-labels), FairNet does not fall below ERM baseline performance.
>
> These findings confirm that FairNet-Unlabeled is not only effective, but also robust—even for outlier samples or noisy signals.
>
> # Response to Q5:
>
> Thank you for this insightful question. While our study focuses on fairness under a fixed distribution, we agree that robustness to distribution shift is critical.
>
> Although we did not explicitly test FairNet under distribution shift, its design may help in such scenarios. In particular:
>
> - The **contrastive loss** promotes class-level representations that are invariant to sensitive attributes, reducing reliance on spurious correlations.
> - The **conditional LoRA module** enables flexible adjustments without overfitting to specific data patterns.
>
> These properties could contribute to better generalization under distribution shift, though we have not yet validated this empirically. We appreciate the suggestion and consider this an important direction for future work.

---

> > ### Comment · Reviewer_niDK · 2025-08-05
> >
> > thank you for your thoughtful response.
> > I have read the author's rebuttal. They addressed my concerns. I like the paper, as it suggests a simple yet effective idea.

---

> > > ### Author Response · Authors · 2025-08-06
> > >
> > > Dear Reviewer,
> > >
> > > Thank you very much for your positive feedback. We are delighted to hear that our rebuttal has satisfactorily addressed your concerns.
> > >
> > > We sincerely appreciate your thoughtful comments and kind support throughout the review process.
> > >
> > > Sincerely,
> > >
> > > The Authors

---

### Note · Authors · 2025-08-13

Dear Area Chair,

We thank you and the reviewers for the thoughtful discussion. This final note focuses on facts most relevant for the decision.

**Core takeaway**. FairNet performs dynamic, instance-level fairness correction by gating Conditional LoRA with a lightweight detector. Corrections apply only when needed, so worst-group performance improves while overall accuracy is preserved; Eq. 8 gives a sufficient TPR/FPR condition for non-decreasing accuracy.

**Evidence consolidated during discussion.**

- **Robustness to detector quality.** Across five seeds the detector is stable; stress tests that inject up to 100% noise show a safety property: performance does not fall below ERM. In the unlabeled setting (LOF), the detector attains TPR 74.2% / FPR 7.7%, improving WGA/EOD at ERM-level accuracy.

- **Thresholding.** We used a single τ=0.5 across datasets; varying τ trades fairness for accuracy in a predictable way and can be tuned at low cost.

- **Efficiency.** Lightweight modules yield small overhead (BERT-base: +0.28% params, +3.7% GFLOPs; ViT: +0.67% params, +8.2% GFLOPs). Conditional activation keeps average latency low.

- **Multi-attribute fairness.** A single multi-label detector can gate all LoRA modules. On HateXplain (race+gender), EOD drops 20.1→9.8 (race) and 12.9→8.2 (gender) with accuracy maintained.

- **Ablations beyond CelebA.**  In additional ablations on MultiNLI and CelebA, removing the detector and/or the contrastive loss degrades WGA/EOD; ERM shows the largest EOD. The custom contrastive loss itself lifts WGA over cross-entropy-only training.

**Methodological & ethical clarifications.**

Contrastive targets are class-conditional majority centroids; one shared detector controls all LoRA modules.

We will replace “identifying biased instances” with “instances predicted to belong to performance-disadvantaged groups” and add an ethics note on disparate treatment vs. equitable outcomes.

**Scope & commitments.** Current datasets are mid-scale and distribution shift was not explicitly tested; we will add the new MultiNLI ablation, expanded CelebA ablation, stability tests (error bars across seeds and robustness to detector noise/thresholds), overhead analysis, and clarified language in the camera-ready; we will release code, configs, and seeds.

We believe these clarifications substantiate a simple, retrofit-ready mechanism that improves worst-group accuracy without sacrificing overall performance. Thank you for your consideration.

---

### Decision · Program_Chairs · 2025-09-17

**Decision:**

Accept (poster)

**Comment:**

This paper proposes FairNet, an in-processing, instance-level fairness method that inserts a lightweight bias detector into a frozen backbone and conditionally activates contrastively trained LoRA adapters only when needed. The authors theoretically show that under a TPR/FPR threshold on the detector, FairNet can improve WGA without decreasing overall accuracy. The authors then empirically show that FairNet variants improve WGA and reduce EOD on three datasets.

**Strengths**

The paper tackles a common problem in algorithmic fairness (improving worst-group performance without harming overall accuracy) using a simple modular mechanism. The framework’s ability to operate with full/partial/unlabeled sensitive attributes, plus low computational overhead, improves its practical applicability. Reviewers were generally happy with the experimental evaluations.

**Initial Weaknesses**

1. 2KFg and niDK raised concerns about the TPR/FPR assumption used in the theoretical results holding in practice.

2. 2KFg and niDK requested additional experimental evaluations, including larger-scale or intersectional analyses, sensitivity to $\tau$, computational overhead, and robustness to noisy/weak detectors.

3. 33v7 raised several issues with the clarity of the paper.

**Rebuttal Period**

The authors addressed the core concerns with substantive additions, including several new experimental results which address the primary concerns in W1 and W2, and various clarity changes/rephrasings which address W3.

**Overall Evaluation**

All reviewers felt that their concerns have been addressed by the rebuttals, and all are leaning towards acceptance. As such, I recommend acceptance. The authors should be sure to integrate the new experimental results and promised clarity improvements to the camera ready.